# INFERENCE-TIME SCALING FOR GENERALIST REWARD MODELING

## ABSTRACT

Reinforcement learning (RL) has been widely adopted in post-training for large language models (LLMs) at scale. Recently, the incentivization of reasoning capabilities in LLMs from RL indicates that proper learning methods could enable effective inference-time scalability. A key challenge of RL is to obtain accurate reward signals for LLMs in various domains beyond verifiable questions or artificial rules. In this work, we investigate how to improve reward modeling (RM) with more inference compute for general queries, i.e. the **inference-time scalability of generalist RM**. For the RM approach, we adopt pointwise generative reward modeling (GRM) to enable flexibility for different input types and the potential for inference-time scaling. For the learning method, we propose **Self-Principled Critique Tuning** (SPCT) to foster scalable reward generation behaviors in GRMs through online RL, to generate principles adaptively and critiques accurately, resulting in **SPCT-GRM** models. Furthermore, for effective inference-time scaling, we use parallel sampling to expand compute usage, and introduce a meta RM to guide the voting process for better scaling performance. Empirically, we show that SPCT significantly improves the quality and scalability of GRMs, outperforming existing methods and models in various RM benchmarks without severe biases, and it can achieve better performance compared to training-time scaling. SPCT-GRM still meets challenges in some tasks, which we believe can be addressed by future efforts in generalist reward systems. The models will be released.

## 1 INTRODUCTION

The remarkable advancements in large language models (LLMs) (DeepSeek-AI, 2024b; OpenAI, 2025b) have catalyzed significant shifts in artificial intelligence research, enabling models to perform tasks that require understanding, generation, and nuanced decision-making capabilities. Recently, reinforcement learning (RL) as a post-training method for LLMs has been widely adopted at scale, and resulting in remarkable improvements in human value alignment (Ouyang et al., 2022; Bai et al., 2022a), long-term reasoning (DeepSeek-AI, 2025; OpenAI, 2025c), and environment adaptation (OpenAI, 2025a) for LLMs. Reward modeling (RM) (Gao et al., 2023), as a crucial component in RL, is essential for generating accurate reward signals for LLM responses. Current studies (Lightman et al., 2024; DeepSeek-AI, 2025) also show that, with high-quality and robust rewards in either training or inference time,
LLMs can achieve strong performance in specific domains.

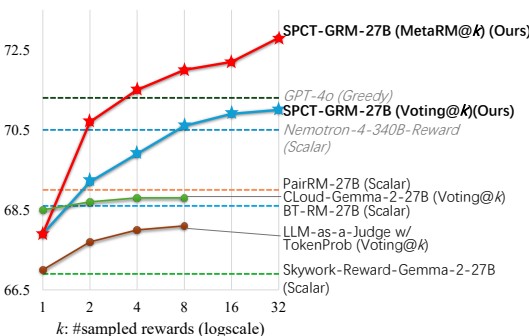

Figure 1: Inference-time scaling performance with different RMs on all tested RM benchmarks. Results are shown with up to 8 samples for each method, and are further scaled to 32 samples for ours. Non-italic font indicates models based on Gemma-2-27B.

However, such high-quality rewards in specific domains are mainly obtained from human-designed environments with clear conditions (Yao et al., 2022; Xie et al., 2024) or from hand-crafted rules for verifiable questions, e.g., mathematical problems (Hendrycks et al., 2021; Veeraboina, 2023) and

coding tasks (Jimenez et al., 2024; Zhuo et al., 2025). In general domains, reward generation is more challenging, as the criteria for rewards are more diverse and complex, and there are often no explicit reference or ground truth. Generalist reward modeling is thus crucial for improving the performance of LLMs in broader applications, either from post-training perspectives, e.g., RL at scale, or from inference perspectives, e.g., RM-guided search. Furthermore, RM performance should be improved by increasing both the training compute (Gao et al., 2023) and the inference compute.

In practice, challenges arise in making RMs both general and effectively scalable in inference time. The former demands (1) flexibility for different input types and (2) accurate reward generation in various domains. We refer to this paradigm as **generalist reward modeling**. Moreover, effective **inference-time scalability** requires the RM (3) to generate higher-quality reward signals with increased inference compute, and (4) to learn scalable behaviors for better performance-compute scaling. Existing research on reward modeling demonstrates several paradigms for reward generation, including scalar (Cobbe et al., 2021; Wang et al., 2024e; Liu et al., 2024), semi-scalar (Ye et al., 2025a; Yu et al., 2025b; Zhang et al., 2025a), and generative (Li et al., 2024a; Kim et al., 2024; Vu et al., 2024; Cao et al., 2024; Arabzadeh et al., 2024; Ye et al., 2025b; Alexandru et al., 2025; Yu et al., 2025a) approaches, and various scoring patterns, such as pointwise (Kendall & Smith, 1940; Gao et al., 2023; Yuan et al., 2024; Winata et al., 2025; Guo et al., 2025) and pairwise (Park et al., 2024; Zheng et al., 2023; Jiang et al., 2023; Wang et al., 2024c; Liu et al., 2025). These approaches inherently determine the input flexibility and the inference-time scalability of RMs (*(1)&(3)*), as shown in Figure 2. For instance, pairwise RMs only consider the relative preference of paired responses, lacking flexibility to accept single or multiple responses as input; scalar RMs could hardly generate diverse reward signals for the same response, which obstructs getting better rewards through sampling-based inference-time scaling methods (Snell et al., 2025). Also, different learning methods (Wang et al., 2024a; Ankner et al., 2024; Wang et al., 2024c; Mahan et al., 2024) have been proposed to improve the quality of rewards, but few of them focus on inference-time scalability and study the interconnection between the learned reward generation behaviors and the effectiveness of inference-time scaling of RMs, resulting in marginal performance improvement (*(2)&(4)*). Current research (DeepSeek-AI, 2025) indicates that effective inference-time scalability could be enabled by proper learning methods, which raises the question: *Can we design a learning method aiming to enable effective inference-time scaling for generalist reward modeling?*

In this work, we investigated different approaches for RM, and found that pointwise generative reward modeling (GRM) could unify the scoring of single, paired, and multiple responses within pure language representation, overcoming challenge *(1)*. We explored that certain principles could guide reward generation within proper criteria for GRMs, improving the quality of rewards, which suggested that *inference-time scalability of RM might be achieved by scaling the generation of high-quality principles and accurate critiques*. Based on this preliminary, we propose a novel learning method, **Self-Principled Critique Tuning** (SPCT), to foster effective inference-time scalable behaviors in GRMs. By leveraging rule-based online RL, SPCT enables GRMs to learn to adaptively posit principles and critiques based on the input query and responses, leading to better outcome rewards in general domains (challenge *(2)*). We then come up with **SPCT-GRM-27B**, which is post-trained with SPCT based on Gemma-2-27B (Team, 2024). For inference-time scaling, we expand compute usage by sampling multiple times. By sampling in parallel, SPCT-GRM could generate different sets of principles and corresponding critiques, and then vote for the final reward. **With larger-scale sampling, SPCT-GRM could judge more accurately based on more diverse principles, and output rewards with finer granularity**, which resolves challenge *(3)&(4)*. Furthermore, we train a meta RM in addition to voting for better scaling performance. Empirically, we show that SPCT significantly improves the quality and scalability of GRMs, outperforming existing methods and models in multiple comprehensive RM benchmarks without severe domain biases. We also compared the inference-time scaling performance of SPCT-GRM-27B with larger models up to 671B parameters, and found it could achieve better performance compared to training-time scaling on model sizes. Though the current method meets challenges in efficiency and specific tasks, with efforts beyond SPCT, we believe GRMs with enhanced scalability and efficiency could serve as a versatile interface for generalist reward systems, advancing the frontiers of LLM post-training and inference.

In general, our main contributions are as follows.

1. We propose a novel approach, **Self-Principled Critique Tuning** (SPCT), to foster effective inference-time scalability for generalist reward modeling, resulting in **SPCT-GRM** models.

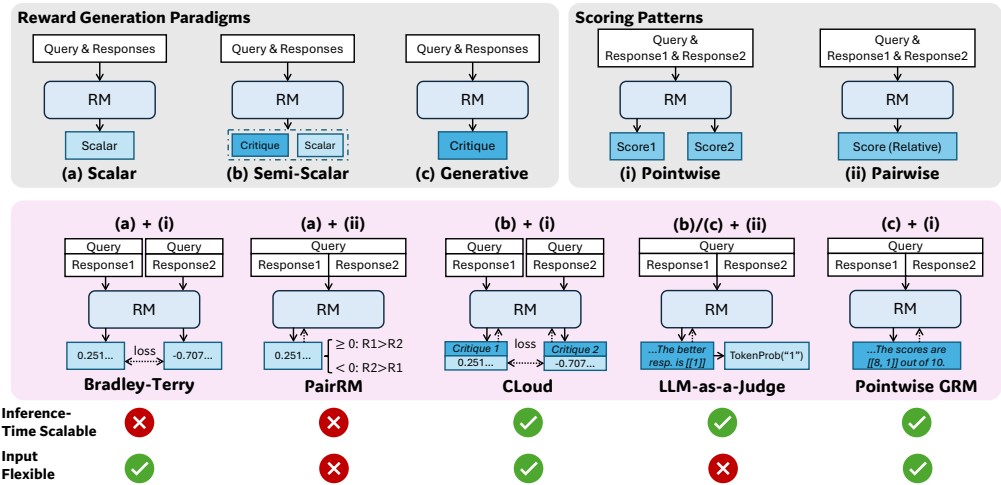

Figure 2: Different paradigms for reward generation, including (a) scalar, (b) semi-scalar, and (c) generative approaches, and different scoring patterns, including (i) pointwise and (ii) pairwise approaches. We list the representative methods for each approach, and corresponding inference-time scalability (whether better rewards could be obtained from multiple sampling) and input flexibility (whether supports rating single and multiple responses).

> And we further introduce a meta RM to effectively improve the inference-time scaling performance of SPCT-GRM beyond voting.
>
> 2. We empirically show SPCT significantly improves the quality and inference-time scalability of GRMs over existing methods and several strong public models.
>
> 3. We also applied the SPCT training schedule to LLMs with larger sizes and found that inference-time scaling could outperform model size scaling in training time.

## 2 PRELIMINARIES

### 2.1 COMPARISONS OF DIFFERENT RM APPROACHES

As shown in Figure 2, RM approaches are mainly determined by reward generation paradigms and scoring patterns, which inherently affect the inference-time scalability and the input flexibility of the RM. For **reward generation paradigms**, we distinguish three main approaches: scalar, semi-scalar, and generative. The scalar approach assigns scalar values to the given query and responses, while the semi-scalar approach generates textual judgement, termed "critique", and the scalar reward value as well. The generative approach only generates critiques as the textual reward, from which the reward value could be extracted. For **scoring patterns**, we distinguish two main approaches: pointwise and pairwise. The pointwise approach assigns an individual score to each response, while the pairwise approach selects a single best response from all candidates.

To expand inference compute usage, we focus on sampling-based methods, which generate multiple sets of rewards for the same query and responses, and then aggregate the final reward. Thus, the *inference-time scalability* of RMs is determined by whether different rewards could be obtained from multiple sampling, where scalar RMs would fail in most cases due to the invariant generation of rewards; and the *input flexibility* is defined by whether the RM supports rating single, paired, and multiple responses, where pairwise RMs could hardly rate single responses and usually require extra techniques (Jiang et al., 2023; Liu et al., 2025) to handle multiple responses. The formulation of pointwise GRMs is:

$$\{S_i\}_{i=1}^n = f_{\text{point}}\left(\mathcal{R}, \{y_i\}_{i=1}^n\right) = f_{\text{extract}}(\boldsymbol{C}), \quad \mathcal{R} = \boldsymbol{C} \sim r_\theta\left(x, \{y_i\}_{i=1}^n\right), \quad (1)$$

where $x$ is the query, $y_i$ is the $i$-th response, $r_\theta$ is the reward function parameterized by $\theta$, $\mathcal{R}$ is the generated reward, $\boldsymbol{C}$ is the textual judgement or critique, $S_i \in \mathbb{R}$ is the individual score of $y_i$, $f_{\text{point}}$ denotes the pointwise scoring function, and $f_{\text{extract}}(\cdot)$ is the implementation of $f_{\text{point}}$ to extract the rewards from textual generation results. Usually, the scores are discrete, and in this work, we assign $S_i \in \mathbb{N}, 1 \le S_i \le 10$ by default. Detailed analysis is in Appendix D.1.

## 2.2 Boosting Reward Quality with Principles

Generalist RM requires generating high-quality rewards beyond specific domains (Hendrycks et al., 2021; Jimenez et al., 2024), where the criteria for rewards are more diverse and complex, and there are often no explicit reference or ground truth. To this end, for general domains, we adopt principles to guide reward generation in place of artificial rules. Principles for LLMs are first introduced in Constitutional AI (Bai et al., 2022b; Sharma et al., 2025), which are hand-crafted criteria that guide the LLMs or curated classifiers to construct safe data pipelines. With principles, the reward generation of GRMs changes to

$$\mathcal{R} = \boldsymbol{C} \sim r_\theta \left( x, \{y_i\}_{i=1}^n, \{p_i\}_{i=1}^m \right), \tag{2}$$

where $\{p_i\}_{i=1}^m$ denotes the principles. We conduct a preliminary experiment to examine the influence of proper principles on reward quality, with the Chat Hard subset of Reward Bench (Lambert et al., 2024) and the IFEval subset of the PPE benchmark (Frick et al., 2025).

In the experiment, data samples contain a query and two responses, with the ground-truth label denoting the better response. We use GPT-4o-2024-08-06 to generate the principles and then pointwise rewards four times for each sample. We filter the principles from the correct reward generation process, where the larger reward value is assigned to the labeled better response. We test different LLMs with principles generated by themselves and the filtered principles, and compare them with the default setting with no principle guidance. The results are shown in Table 1. We found that the self-generated principles barely improve performance, but the filtered principles could significantly boost the reward quality. The result is non-trivial and two main conclusions could be drawn: (a) Current LLMs could generate diverse principles, but not all of them are proper for reward generation. (b) A subset of generated principles could better guide reward generation under correct criteria, indicating a potential of self-bootstrapping. These findings are the foundation of the usage of online RL to optimize GRMs, where they could learn from the principles generated by themselves with a clear signal to tell whether the principles are proper or not. Other details are depicted in Appendix E.

Table 1: Preliminary experiments on the influence of principles on reward quality. The default setting of SPCT-GRM-27B includes self-generated principles.

| Method | Chat Hard | IFEval |
|---|---|---|
| **GPT-4o-2024-08-06** | 76.1 | 56.0 |
| w/ Self-Gen. Principles | 75.9 | 55.6 |
| w/ Filtered Principles | 77.8 | 57.5 |
| **Gemma-2-27B-it** | 59.1 | 56.1 |
| w/ Self-Gen. Principles | 64.0 | 55.8 |
| w/ Filtered Principles | 68.0 | 57.3 |

## 3 Self-Principled Critique Tuning (SPCT)

Inspired by the preliminary results, we developed a novel approach for pointwise GRMs to learn generating adaptive and high-quality principles that could effectively guide the generation of critiques, termed **Self-Principled Critique Tuning** (SPCT). As shown in Figure 3, SPCT consists of two phases: rejective fine-tuning, as the cold start, and rule-based online RL, reinforcing generalist reward generation by advancing the generated principles and critiques. SPCT fosters these behaviors in GRMs for inference-time scaling as well.

### 3.1 Unpinning Principles from Understanding to Generation

From preliminary experiments in Section 2.2, we found that proper principles could guide reward generation within certain criteria, which is critical for high-quality rewards. However, it remains challenging to generate effective principles for generalist RM at scale. To address this challenge, we propose to unpin principles from understanding to generation, i.e. view principles as a part of reward generation instead of a preprocessing step.

Formally, principles guide the generation of rewards following Equation 2, when principles are pre-defined. GRMs could generate principles themselves, and then generate critiques based on the principles, formalized as

$$\{p_i\}_{i=1}^m \sim p_\theta \left( x, \{y_i\}_{i=1}^n \right), \quad \mathcal{R} = \boldsymbol{C} \sim r_\theta \left( x, \{y_i\}_{i=1}^n, \{p_i\}_{i=1}^m \right), \tag{3}$$

where $p_\theta$ is the principle generation function parameterized by $\theta$, that shares the same model with reward generation $r_\theta$. In practice, they are implemented with the same language head. This shift enables principles to be generated based on the input query and responses, adaptively aligning reward generation process. The quality and granularity of the principles and corresponding critiques could be improved with post-training on GRMs. With **the principles generated at scale**, GRMs could rate rewards with finer granularity and broader criteria to achieve better inference-time scalability.

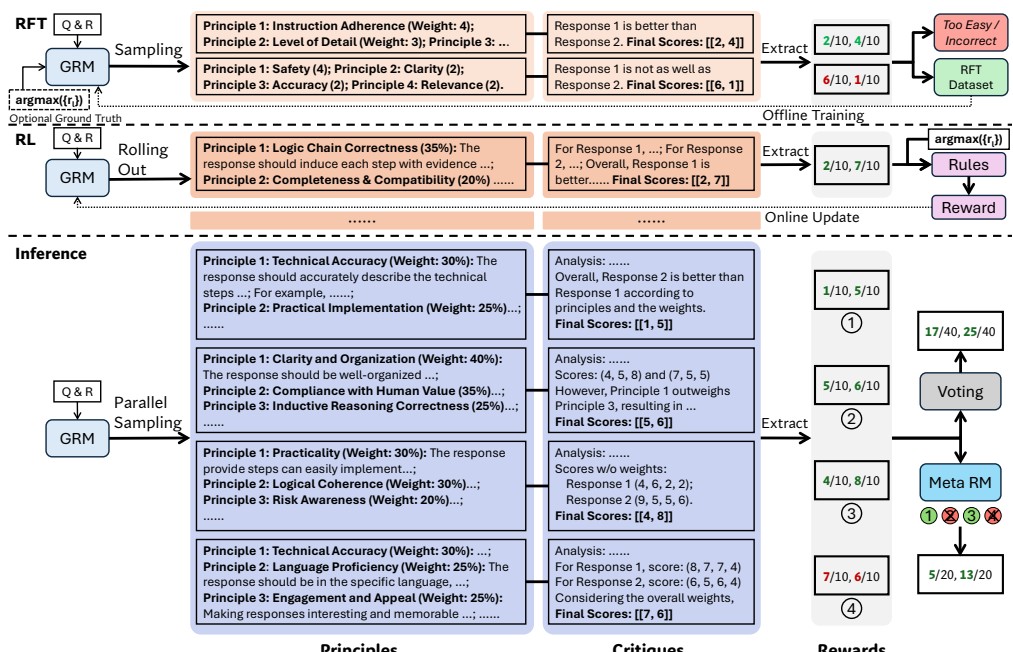

Figure 3: Illustration of SPCT, including rejective fine-tuning, rule-based RL, and scalable behaviors during inference. The inference-time scaling is achieved via voting or meta RM guided voting with principles generated at scale, resulting in finer-grained outcome rewards within a larger value space.

## 3.2 Rule-Based Reinforcement Learning

To optimize principle and critique generation in GRMs simultaneously, we propose SPCT, which integrates rejective fine-tuning and rule-based RL. The former serves as a cold start.

**Rejective Fine-Tuning (Cold Start)** The core idea of the rejective fine-tuning stage is to train the GRM to generate principles and critiques in the correct format and for various input types. Unlike previous works (Vu et al., 2024; Cao et al., 2024; Alexandru et al., 2025) that mix RM data for single, paired, and multiple responses in different formats, we adopt pointwise GRM, introduced in Section 2.1, to flexibly generate rewards for any number of responses in the same format. For data construction, besides general instruction data, we sample trajectories with a pretrained GRM by giving the query and corresponding responses. Each RM data point contains a query and one or multiple responses to the query, as well as the ground-truth label denoting the best response. For each RM data point, the sampling of principles and critiques is performed $N_{\text{RFT}}$ times. The rejection strategy is also unified, which is to reject trajectories with predicted rewards that are incorrect, and the query and responses with all $N_{\text{RFT}}$ trajectories correct (too easy). Formally, let $r_i$ denote the ground-truth reward for the $i$-th response $y_i$ to the query $x$, the predicted pointwise rewards $\{S_i\}_{i=1}^n$ are correct if

$$\begin{cases} \forall i \neq j, \quad S_j > S_i, \quad j = \arg\max_l\{r_l\}_{l=1}^n, & \text{if } n \geq 2, \\ S_1 = r_1, & \text{if } n = 1. \end{cases} \tag{4}$$

with a guarantee that the ground-truth rewards only contain one maximum. However, similar to previous works (Zhang et al., 2025a), we found pretrained GRMs could hardly generate correct rewards for a portion of data points within limited sampling quota. Thus, we optionally append $\arg\max_l\{r_l\}_{l=1}^n$ to the prompt of the GRM, termed *hinted sampling*, with the expectation that the predicted rewards to align with the ground truth, besides *non-hinted sampling*. Specifically, an additional segment "The best response is: Response $\arg\max_l\{r_l\}_{l=1}^n$" will be appended to the input. For hinted sampling, each query and the corresponding responses are sampled once, and trajectories are only rejected when incorrect. Beyond previous studies (Li et al., 2024a; Mahan et al., 2024), we observed that hinted sampled trajectories sometimes take shortcuts in judgement, especially for reasoning tasks, indicating the necessity and potential benefits of non-hinted sampled data.

**Rule-Based RL** The GRM is further fine-tuned with rule-based online RL. Specifically, we use the original setting of GRPO (Shao et al., 2024) with rule-based outcome rewards. During roll-

out, the GRM generates principles and critiques based on the input query and responses, and then the predicted reward is extracted and compared to the ground truth with accuracy rules. Unlike DeepSeek-AI (2025), no format rewards are used. Instead, a larger coefficient for the KL penalty is applied to ensure the format and avoid severe biases. Formally, the reward for the $i$-th output $o_i$ to the given query $x$ and responses $\{y_i\}_{i=1}^n$ is

$$\hat{r}_i = \begin{cases} 1, & \text{if } n \geq 2 \text{ and } \forall i' \neq j', \quad S_{j'} > S_{i'}, \quad j' = \arg\max_l \{r_l\}_{l=1}^n, \\ 1, & \text{if } n = 1 \text{ and } S_1 = r_1, \\ -1, & \text{otherwise}, \end{cases} \tag{5}$$

where the pointwise rewards $\{S_i\}_{i=1}^n$ are extracted from $o_i$. **The reward function encourages GRMs to distinguish the best responses with online optimized principles and critiques, in favor of effective inference-time scaling.** The reward signal could be obtained seamlessly from any preference dataset and labeled LLM responses.

## 4 INFERENCE-TIME SCALING WITH SPCT

To further improve the performance of SPCT-GRM using more inference compute, we explore sampling-based strategies to achieve effective inference-time scalability.

**Voting with Generated Rewards** Voting is a widely adopted method for inference-time scaling in RM. Recalling the approaches in Section 2.1, we demonstrate voting results of $k$ samples for semi-scalar and generative RMs. For semi-scalar RMs (Ankner et al., 2024; Zhang et al., 2025a), voting is performed as averaging:

$$\boldsymbol{S}^* = \frac{1}{k}\sum_{i=1}^k \boldsymbol{S}_i, \quad \{\mathcal{R}_i = (\boldsymbol{S}_i, \boldsymbol{C}_i)\}_{i=1}^k \sim r_\theta\left(x, \{y_i\}_{i=1}^n\right), \tag{6}$$

where $\boldsymbol{S}^*$ is the final reward. In practice, the scalar value has limited variance which could hinder the scalability. For pairwise GRMs (Mahan et al., 2024; Wang et al., 2024c), voting is performed as selecting the response identified to be the best with the highest frequency, i.e. majority:

$$\hat{y}^* = \arg\max_y \sum_{i=1}^k \mathbb{I}(y = \hat{y}_i), \quad \{\mathcal{R}_i = \boldsymbol{C}_i\}_{i=1}^k \sim r_\theta\left(x, \{y_i\}_{i=1}^n\right), \tag{7}$$

where $\hat{y}^*$ is the predicted best response, $f_{\text{pair}}(\cdot, \cdot)$ is a selection function, $\hat{y}_i = f_{\text{pair}}(\boldsymbol{C}_i, \{y_i\}_{i=1}^n)$ is the individually selected best response of each sample, and $\mathbb{I}(\cdot)$ is the indicator function. Though the voting process is scalable, the majority voted result might be biased since ties are not allowed in each sample, and may not be able to tell apart subtle differences between responses due to the lack of quantitative scores. The voting process for pointwise GRMs is defined as summing the rewards:

$$S_i^* = \sum_{j=1}^k S_{i,j}, \quad \{p_{i,j}\}_{i=1}^{m_j} \sim p_\theta\left(x, \{y_i\}_{i=1}^n\right), \mathcal{R}_j = \boldsymbol{C}_j \sim r_\theta\left(x, \{y_i\}_{i=1}^n, \{p_{i,j}\}_{i=1}^{m_j}\right), j = 1, ..., k, \tag{8}$$

where $S_i^*$ is the final reward for the $i$-th response ($i = 1, ..., n$) and $\{S_{i,j}\}_{i=1}^n = f_{\text{point}}(\boldsymbol{C}_j, \{y_i\}_{i=1}^n)$ is the $j$-th set of pointwise rewards. **Since $S_{i,j}$ is usually set within a small discrete range,** e.g., $\{1, ..., 10\}$, **the voting process actually expands the reward space by $k$ times, and enables the GRM to generate a large number of principles, which benefits the quality and granularity of the final rewards.** An intuitive explanation is that, if each principle could be viewed as a proxy of judgement perspectives, a larger number of principles may reflect the real distribution more accurately, leading to scaling effectiveness. Notably, to avoid positional biases and for diversity, responses are shuffled before sampling.

**Meta Reward Modeling Guided Voting** The voting process of SPCT-GRM requires multiple sampling and a few generated principles and critiques might be biased or low-quality due to randomness or model limitations. Thus, we train a meta RM to guide the voting process. The meta RM is a pointwise scalar RM, trained to identify the correctness of the principles and critiques generated by SPCT-GRM, with the binary cross-entropy loss, where the label is identified based on Equation 4. The prompt template is in Appendix H, integrates the query, candidate responses, corresponding principles, and the critiques. The dataset comprises trajectories from non-hinted sampling in the RFT stage, and trajectories sampled from the SPCT-GRM, to provide positive and negative rewards and alleviate the gap between training and inference policy (Chow et al., 2025). The guided voting is simple: The meta RM outputs meta rewards for $k$ sampled rewards, and the final outcome is voted by rewards with top $k_{\text{meta}} \leq k$ meta rewards, so that filtering out low-quality samples.

Table 2: Overall results of different methods and models on RM benchmarks. Underlined numbers indicate the best performance, **bold numbers** indicate the best performance among baseline and our methods, and *italicized font* denotes scalar or semi-scalar RMs. For meta RM guided voting (MetaRM), $k_{\text{meta}} = \frac{1}{2}k$. "Avg. Rank ($\downarrow$)" is calculated based on the rank (from 1 to 6, better ranks denoted by darker shades) of baseline and our methods on each dataset.

| Model | RB | PPE Pref. | PPE Correct. | RMB | Overall | Avg. Rank ($\downarrow$) |
|---|---|---|---|---|---|---|
| *Reported Results of Public Models* | | | | | | |
| *Skywork-Reward-Gemma-2-27B* | 94.1 | 56.6 | 56.6 | 60.2 | 66.9 | - |
| DeepSeek-V2.5-0905 | 81.5 | 62.8 | 58.5 | 65.7 | 67.1 | - |
| Gemini-1.5-Pro | 86.8 | 66.1 | 59.8 | 56.5 | 67.3 | - |
| *ArmoRM-8B-v0.1* | 90.4 | 60.6 | 61.2 | 64.6 | 69.2 | - |
| *InternLM2-20B-Reward* | 90.2 | 61.0 | 63.0 | 62.9 | 69.3 | - |
| LLaMA-3.1-70b-Instruct | 84.1 | 65.3 | 59.2 | 68.9 | 69.4 | - |
| Claude-3.5-sonnet | 84.2 | 65.3 | 58.8 | 70.6 | 69.7 | - |
| *Nemotron-4-340B-Reward* | 92.0 | 59.3 | 60.8 | 69.9 | 70.5 | - |
| GPT-4o | 86.7 | 67.1 | 57.6 | 73.8 | 71.3 | - |
| *Reproduced Results of Baseline Methods* | | | | | | |
| **LLM-as-a-Judge** | 83.4 | 64.2 | 58.8 | 64.8 | 67.8 | 4.50 |
| **BT-RM-27B** | *81.7* | **68.3** | **66.7** | *57.9* | 68.6 | 3.50 |
| **CLoud-Gemma-2-27B** | *82.0* | 67.1 | 62.4 | 63.4 | 68.7 | 3.50 |
| **PairRM-27B** | 87.1 | 65.8 | 64.8 | 58.2 | 69.0 | **2.75** |
| *Results of Our Method* | | | | | | |
| **SPCT-GRM-27B-RFT (Ours)** | 84.5 | 64.1 | 59.6 | 67.0 | 68.8 | 4.00 |
| **SPCT-GRM-27B (Ours)** | 86.0 | 64.7 | 59.8 | 69.0 | 69.9 | **2.75** |
| *Results of Inference-Time Scaling (Voting@32)* | | | | | | |
| **SPCT-GRM-27B (Ours)** | 88.5 | 65.3 | 60.4 | 69.7 | 71.0 | - |
| **SPCT-GRM-27B (MetaRM) (Ours)** | 90.4 | 67.2 | 63.2 | **70.3** | 72.8 | - |

# 5 RESULTS ON REWARD MODELING BENCHMARKS

## 5.1 EXPERIMENT SETTINGS

**Benchmarks and Evaluation Metrics** We evaluate the performance of different methods on various RM benchmarks of different domains: **Reward Bench (RB)** (Lambert et al., 2024), **PPE** (Preference and Correctness subsets) (Frick et al., 2025), **RMB** (Zhou et al., 2025), **ReaLMistake** (Kamoi et al., 2024). We use the standard evaluation metrics for each benchmark: accuracy of picking the best response from a set of responses in Reward Bench, PPE, and RMB, and ROC-AUC for ReaLMistake. To deal with ties of the predicted rewards for multiple responses, we shuffle the responses and determine the best response by $\arg\max_i S_i$, where $S_i$ is the predicted reward for the $i$-th response after shuffling. Details are in Appendix E.

**Method Implementation** For the baseline methods, we re-implement **LLM-as-a-Judge** (Zheng et al., 2023), **BT-RM-27B** (Bradley-Terry model) (Kendall & Smith, 1940), **CLoud-Gemma-2-27B** (Ankner et al., 2024), and **PairRM-27B** (Jiang et al., 2023) based on Gemma-2-27B (Team, 2024) and with all compatible training data and settings as SPCT-GRM. For our methods, we implement **SPCT-GRM-27B-RFT** based on Gemma-2-27B, and **SPCT-GRM** on different sizes of LLMs, including DeepSeek-V2-Lite (16B MoE) (DeepSeek-AI, 2024a), Gemma-2-27B, DeepSeek-V2.5 (236B MoE), and DeepSeek-V3 (671B MoE) (DeepSeek-AI, 2024b). The meta RM is trained on Gemma-2-27B. Default results are reported with greedy decoding, and the inference-time scaling uses $\text{temperature} = 0.5$. Other details are provided in Appendix D.

## 5.2 RESULTS AND ANALYSIS

**Performance on RM Benchmarks** The overall results of different methods and models on RM benchmarks are shown in Table 2. We compare the performance of SPCT-GRM-27B with the reported results of public models and the reproduced results of baseline methods. We find that SPCT-GRM-27B outperforms the baseline methods in overall performance, and achieves competitive performance with strong public RMs, such as Nemotron-4-340B-Reward and GPT-4o; with inference-time scaling, SPCT-GRM-27B could further improve and achieve the best overall results. For detailed comparisons, scalar (BT-RM-27B) and semi-scalar (CLoud-Gemma-2-27B) RMs demonstrate biased results on different benchmarks, with significantly better performance on verifiable tasks (PPE Correctness) than all generative RMs, but fail in different other benchmarks,

Table 3: Inference-time scalability results of different methods on RM benchmarks. Settings are the same as Table 2.

| Model | Overall |
|---|---|
| *Reported Results of Public Models* | |
| *Nemotron-4-340B-Reward* | *70.5* |
| GPT-4o | 71.3 |
| *Results of Inference-Time Scaling (Voting@1)* | |
| **LLM-as-a-Judge** | 67.0 |
| *CLoud-Gemma-2-27B* | *68.5* |
| **SPCT-GRM-27B-RFT (Ours)** | 67.8 |
| **SPCT-GRM-27B (Ours)** | 67.9 |
| *Results of Inference-Time Scaling (Voting@8)* | |
| **LLM-as-a-Judge** | 67.6 (+0.6) |
| **LLM-as-a-Judge w/ TokenProb** | 68.1 (+1.1) |
| *CLoud-Gemma-2-27B* | *68.8 (+0.3)* |
| **SPCT-GRM-27B-RFT (Ours)** | 69.3 (+1.5) |
| **SPCT-GRM-27B (Ours)** | 70.6 (+2.7) |
| **SPCT-GRM-27B (MetaRM) (Ours)** | 72.0 (+4.1) |
| *Results of Further Inference-Time Scaling (Voting@32)* | |
| **SPCT-GRM-27B (Ours)** | 71.0 (+3.1) |
| **SPCT-GRM-27B (MetaRM) (Ours)** | 72.8 (+4.9) |

Table 4: Ablation studies for different components of the proposed SPCT. **Bold numbers** indicate the best performance.

| Method | Overall |
|---|---|
| *Results of Greedy Decoding* | |
| **SPCT-GRM-27B** | **69.9** |
| w/o Principle Generation | 67.5 |
| w/o Rejective Sampling | 68.7 |
| **SPCT-GRM-27B-RFT** | **68.8** |
| w/o Hinted Sampling (①) | 68.0 |
| w/o Non-Hinted Sampling (②) | 67.4 |
| w/o Rejective Sampling (①&②) | 66.1 |
| w/o General Instruction Data | 63.3 |
| *Results of Inference-Time Scaling (Voting@8)* | |
| **SPCT-GRM-27B** | **70.6** |
| w/o Principle Generation | 68.0 |
| *Results of Inference-Time Scaling (Voting@32)* | |
| **SPCT-GRM-27B** | 71.0 |
| **SPCT-GRM-27B** ($k_{\mathrm{meta}} = 1$) | 71.5 |
| **SPCT-GRM-27B** ($k_{\mathrm{meta}} = 8$) | 72.7 |
| **SPCT-GRM-27B** ($k_{\mathrm{meta}} = 16$) | **72.8** |

respectively. Nonetheless, most public scalar RMs also exhibit severe domain biases. The PairRM approach could alleviate the problem. LLM-as-a-Judge shows similar trends with SPCT-GRM-27B with lower performance, potentially due to the lack of training on rating single responses. In conclusion, **SPCT improves the generalist reward generation capability of GRMs, with significantly fewer biases compared to scalar and semi-scalar RMs**.

**Inference-Time Scalability** The inference-time scaling results of different methods are shown in Table 3, and the overall trends are demonstrated in Figure 1. Details are in Appendix E.3. With up to 8 samples, we find that SPCT-GRM-27B has the highest performance increase over the greedy decoding and sampling results. SPCT-GRM-27B further shows a strong potential to increase performance with larger inference compute, up to 32 samples. We attribute the effectiveness to the refined principle generation, which expands the output length in a structural way and guides the outcome rewards closer to the ground-truth distribution. The meta RM also reveals its validity in filtering low-quality trajectories for SPCT-GRM on each benchmark. Voted with token probabilities, LLM-as-a-Judge also shows a significant performance increase, indicating that the token probability as quantitative weights could help the reliability over mere majority voting on discrete indices. For CLoud-Gemma-2-27B, the performance increase is limited, mainly due to the lack of variance in scalar reward generation, even though the critique has changed a lot. In summary, **SPCT improves the inference-time scalability of GRMs, and the meta RM further boosts the scaling performance in general**.

**Ablation Study** Table 4 shows the ablation study results of different components of the proposed SPCT, detailed results are listed in Appendix E.3. Surprisingly, without the cold start with rejective sampled critique data, general-instruction-tuned GRMs still improve significantly after undergoing the online RL ($66.1 \rightarrow 68.7$). Also, the non-hinted sampling seems more important than the hinted sampling, potentially because of the shortcuts appearing in hinted sampled trajectories. These indicate the **importance of online training for GRMs**. Aligned with previous works (Cao et al., 2024), we confirm that the general instruction data is essential for the performance of GRMs. We find that **the principle generation is crucial for the performance of both greedy decoding and inference-time scaling of SPCT-GRM-27B**. For inference-time scaling, the meta RM guided voting shows robustness with different $k_{\mathrm{meta}}$. Further analysis on the generalist RM performance, including input flexibility, domain generalization of training data, etc., is discussed in Appendix F.

**Scaling Inference and Training Costs** We further investigate the inference-time and training-time scaling performance of SPCT-GRM-27B, by post-training LLMs of different sizes. The models are tested on the Reward Bench, and the results are shown in Figure 4. We find that direct voting with 32 samples of SPCT-GRM-27B could achieve comparable performance to the 671B MoE model, and the meta RM guided voting could achieve the best results with 8 samples, demonstrating the **effectiveness of inference-time scaling of SPCT-GRM-27B compared to scaling model sizes**. Moreover, we test DeepSeek-R1-0120 with a downsampled test set containing 300 samples, and find that

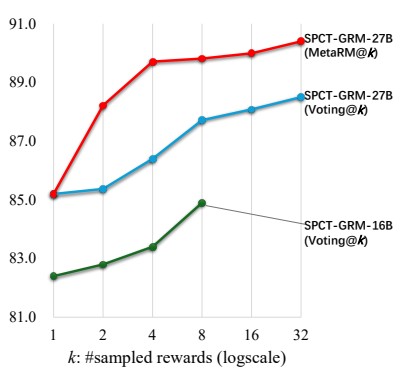 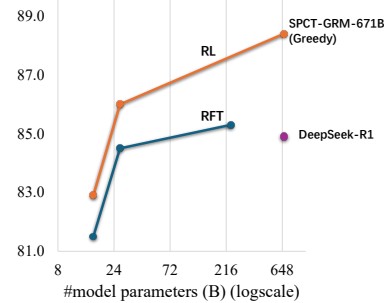

(a) Inference-time scaling results of SPCT-GRM-16B and SPCT-GRM-27B ($k_{\text{meta}} = \frac{1}{2}k$).

(b) Training-time scaling results with different model sizes, using greedy decoding except DeepSeek-R1.

Figure 4: Inference-time scaling v.s. training-time scaling performance on Reward Bench.

its performance is even worse than the 236B MoE RFT model, indicating that expanding long chain-of-thoughts for reasoning tasks could not significantly improve the performance of generalist RM.

## 6 RELATED WORK

**Generative Reward Models** GRMs represent a paradigm shift from scalar RMs (Ouyang et al., 2022), modeling reward as textual feedback or scores. (Li et al., 2024a; Kim et al., 2024; Wang et al., 2024c; Cao et al., 2024; Vu et al., 2024; Alexandru et al., 2025), enabling richer reward representations and more flexible for judging single and multiple responses. Previously, the LLM-as-a-judge method (Zheng et al., 2023; Wang et al., 2025a) accommodates reference-based or reference-free pairwise judgement for evaluating LLMs. Recent studies use offline and online RL to train GRMs (Wu et al., 2024; Mahan et al., 2024; Yu et al., 2025a; Ye et al., 2025b; Chen et al., 2025), incorporate tools and external knowledge with GRMs (Li et al., 2024b; Peng et al., 2025), and even train GRMs as an interface to adjust rewards from environments (Baker et al., 2025). Very recently, GRMs are also used in productive systems for RL (OpenAI, 2025d). Though these methods face challenges in efficiency, they demonstrate the potential to improve rewards at scale.

**Inference-Time Scaling for LLMs** Inference-time scaling for LLMs has been a critical research direction parallel with scaling LLMs in training time. Studies focus on sampling and RM guided aggregation (Lightman et al., 2024; Brown et al., 2024; Snell et al., 2025; Wu et al., 2025). Recently, long-horizon chain-of-thoughts (Wei et al., 2022) incentivized from LLMs significantly improve the reasoning capabilities of the models in both solving (OpenAI, 2024; DeepSeek-AI, 2025; OpenAI, 2025c) and judging (Zheng et al., 2025; Kim et al., 2025) difficult verifiable questions, as another format of inference-time scaling. However, we do not find an effective way to incentivize long-horizon reward generation for generalist reward modeling with general RM data as DeepSeek-AI (2025), and we leave the combination of reasoning and principle-guided reward generation in a single model for future engineering efforts. There is also research using scalable rewards or verifiers to improve the performance of policy models, in domains of coding (Chen et al., 2023), reasoning (Lifshitz et al., 2025), etc. Thus, the development of inference-time scalable generalist RMs in this work might also contribute to the general performance of policy models by inference-time co-scaling.

## 7 CONCLUSION AND FUTURE WORK

We introduced Self-Principled Critique Tuning (SPCT), a method that enhances the scalability of inference time for generalist reward modeling. With rule-based online RL, SPCT enables adaptive generation of principles and critiques, significantly boosting reward quality and inference-time scalability for GRMs in diverse domains. Empirical results demonstrate that SPCT-GRM surpasses baseline methods and a few strong public RMs, and shows notable improvement through inference-time scaling, particularly with the guidance of the meta RM. Future directions could include integrating GRMs into online RL pipelines as versatile interfaces of reward systems, exploring inference-time co-scaling with policy models, or serving as robust offline evaluators for foundation models.

## ETHICS STATEMENT

Our proposed method, Self-Principled Critique Tuning (SPCT), aims to enhance inference-time scalability of generative reward models (GRMs) for general domains. While this advancement promotes accuracy and consistency in reward modeling, several ethical implications might warrant explicit consideration.

Firstly, even though through our empirical analysis that SPCT-GRM shows fewer biases on different domains, the automated generation of principles and critiques can inadvertently perpetuate or amplify biases when the training data is toxic. We argue that further investigation in the meta RM and other bias mitigation strategies should be prioritized to ensure equitable outcomes. Also, our approach does not aim to diminish human oversight. Instead, we advocate maintaining human-in-the-loop frameworks, and developing reliable proxy methods, like SPCT, to scale human oversight more efficiently and effectively.

Secondly, expanded applicability of the inference-time scalable GRMs across diverse domains might raise concerns regarding transparency, accountability, etc. Since the reward generation behavior is largely emerged from self-bootstrapping, the potential for unfaithful principles and critiques is non-negligible. We demonstrate case studies in Appendix G.1 and limitations in Appendix C, and open-source the model under public supervision, which is essential for maintaining trust and ensuring responsible deployment of the artifact.

Finally, robust validation and ongoing vigilance across varied RM benchmarks and practical scenarios remain crucial. Ethical use of SPCT-GRM necessitates proactive management of risks and continuous evaluation against biases, requiring efforts in research about RM evaluation.

## REPRODUCIBILITY STATEMENT

We have made several efforts to ensure the reproducibility of our work. First of all, we will open-source the models for research purposes and public use. Due to the size of the models, they could not be directly uploaded to the supplementary materials. Although we could not release the in-house training data due to privacy concerns, we have particularly described the open-sourced data sources and statistics in Section 5.1 and Appendix D. Following the training recipe (Section 3), one is able to reproduce the training of SPCT-GRM with open-source training infrastructure, e.g., verl (Sheng et al., 2025), and open-source LLMs, e.g., DeepSeek-V3-Base (DeepSeek-AI, 2024b). With replicated models, one should see the similar trends of the experimental results, including (1) SPCT-GRM has fewer domain biases in reward generation and outperforms other RM approaches; (2) SPCT-GRM shows significant improvement with inference-time scaling, especially with the meta RM; (3) The improvement from inference-time scaling is comparable or better to scaling model sizes in SPCT. Further improvements over the work could be and have been made in the community.

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

CONTENTS

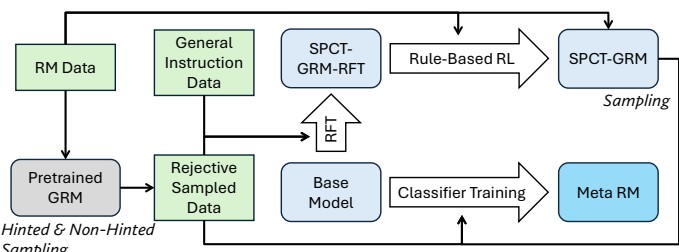

Figure 5: Illustration of the derivation of SPCT-GRM-RFT, SPCT-GRM, and Meta RM in the SPCT pipeline.

## A  THE USE OF LARGE LANGUAGE MODELS (LLMs)

We promise that all the LLM generated content in this work is solely used as research objectives or tools to polish writings in grammar or sentence levels. We do not use LLMs to generate any part of the scientific content from scratch, including but not limited to the ideas, algorithms, experimental results, and analysis. We human authors will take full responsibility for the presented work.

## B  ADDITIONAL RELATED WORK

**Constitutional AI**  Constitutional AI has emerged as a scalable alternative to traditional reinforcement learning from human feedback (Ouyang et al., 2022), aiming to align language models with human values through a set of guiding principles or "constitutions" (Bai et al., 2022b; Sun et al., 2023; 2024), replacing human critiques with AI-generated feedback (Fränken et al., 2024) or classifiers (Sharma et al., 2025) based on these hand-crafted principles. Similarly, rule-based approaches like Sparrow (Glaese et al., 2022) and Rule-Based Rewards (RBR) (Mu et al., 2024) incorporate explicit natural language rules into the training loop for specific domains like safety. Although effective, these methods rely on static, manually written constitutions that are limited in scope, potentially biased, and inflexible. This has motivated interests in automating the generation or refinement of principles, which aligns with our target in this work.

**Scalar Reward Models**  Scalar reward modeling for LLMs are proposed the earliest to serve as a proxy model for human feedback (Stiennon et al., 2020; Gao et al., 2023). Recent studies focus on Bradley-Terry modeling (Kendall & Smith, 1940) and other regression approaches for better expressiveness for scalar reward models (Cai et al., 2024; Wang et al., 2024e;a; Liu et al., 2024; Wang et al., 2025b) of general preference. In contrast to these outcome reward models, process reward models are proposed as step verifiers for reasoning problems, e.g., math, etc. (Cobbe et al., 2021; Wang et al., 2024b; Zhang et al., 2025b), demonstrating the feasibility of scalar RMs in a formal domain with extensive reasoning and knowledge. Scalar RM excels in simplicity and is computationally efficient, but suffers from limited expressivity and struggles to generalize across diverse input types or refine reward signals at inference time.

**Semi-Scalar Reward Models**  Semi-scalar reward models aim to enrich scalar reward signals through textual intermediate representations. (Ye et al., 2025a; Ankner et al., 2024) Consequently, works (Yu et al., 2025b) proposed to enhance the quality of generated critiques to eventually improve reward generation. Some studies use the token probability to substitute the scalar head for reward extraction (Mahan et al., 2024; Zhang et al., 2025a). These works show that semi-scalar RMs face challenges in inference-time scaling based on sampling and voting, resulting in limited performance improvement. The semi-scalar approach trades off between scalar RMs and GRMs in terms of both efficiency and effectiveness.

## C  LIMITATIONS AND FUTURE DIRECTIONS

**Limitations**  Though SPCT significantly leverages the performance and inference-time scalability of GRMs and surpasses (semi-)scalar RMs in general domains, it still faces a few limitations. (1)

The efficiency of the generative RMs is largely lagging behind the scalar RMs at the same scale by nature, which inhibits its large-scale usage in online RL pipelines. However, since we adopt parallel sampling for inference-time scaling, the latency of reward generation with a reasonable amount of, e.g., eight samplings will not increase significantly. Further research around the efficient generation of LLMs and innovations in RM applications could alleviate the problem. (2) In specific domains such as verifiable tasks, SPCT-GRM still lags behind scalar models. This could be because the scalar RMs capture hidden features of reasoning queries and responses, while GRMs need stronger reasoning capabilities to examine responses thoroughly. However, scalar RMs suffer severe biases and scalability issues. For GRMs, we found that both reference-based reward generation (Appendix F.1.3 and long-horizon reasoning (Appendix E.3) could mitigate this limitation. (3) Due to the universality of the pointwise GRM approach, SPCT-GRM could potentially serve as a process RM in addition to the outcome RM. Though we have not explored much in this direction in the paper, the performance in the Reasoning subset of Reward Bench, which mainly comprises of MATH-prm data (Lightman et al., 2024), could partially support the potential of this application.

**Future Directions** There are also several promising directions for future research based on SPCT or SPCT-GRM models. (1) Tool incorporation of RMs is studied by previous work (Li et al., 2024b), and could also be used for SPCT-GRM augmentation. **With tools such as code interpreters and search engine interfaces**, the generated critiques could be more accurate for tasks that requires strict procedures or extensive knowledge, and the cases in which GRMs fail to follow principles related to numeric calculations, pattern matching, etc. could be avoided. (2) **The generation paradigm for principles and critiques could be decomposed** into separate stages, that is, the principles could be generated ahead of time for each query and the responses to be rated and stored, and then the critiques are generated with GRMs, rules, or other agentic approaches. The principle generation serves as an interface for the following critiques. This might improve the efficiency of current GRMs for the integration of RL pipelines. (3) The SPCT-GRM could be potentially **used in LLM offline evaluation**. Since each principle reflects a criteria, we can get criteria from all data points that a particular LLM is inferior than one another, as a interpretable protocol of the weaknesses of the particular LLM. (4) The SPCT-GRM might **be benefit from long-horizon reasoning**. However, this will further affect its efficiency. These directions should be studied in the future work.

## D  IMPLEMENTATION DETAILS

### D.1  COMPARISONS OF DIFFERENT RM APPROACHES

**Reward Generation Paradigms** Classic RMs adopt the **(a) scalar** approach to generate rewards ($\mathcal{R}$), which assigns scalar values to the given query and responses. The scalar approach is further extended to the **(b) semi-scalar** approach, which generates texts besides the scalar value. And the **(c) generative** approach only generates textual rewards.

$$\mathcal{R} = \begin{cases} \boldsymbol{S} & \text{(Scalar)} \\ (\boldsymbol{S}, \boldsymbol{C}) & \text{(Semi-Scalar)} \sim r_\theta\left(x, \{y_i\}_{i=1}^n\right), \\ \boldsymbol{C} & \text{(Generative)} \end{cases} \tag{9}$$

where $x$ is the query, $y_i$ is the $i$-th response, $r_\theta$ is the reward function parameterized by $\theta$, $\boldsymbol{S} \in \mathbb{R}^m, m \leq n$ is the scalar reward, and $\boldsymbol{C}$ is the critique.

**Scoring Patterns** We distinguish two main scoring approaches for rewards: pointwise and pairwise. The **(i) pointwise** approach assigns an individual score to each response:

$$\{S_i\}_{i=1}^n = f_{\text{point}}\left(\mathcal{R}, \{y_i\}_{i=1}^n\right), \quad \mathcal{R} \sim r_\theta\left(x, \{y_i\}_{i=1}^n\right), S_i \in \mathbb{R}, \tag{10}$$

where $f_{\text{point}}(\cdot, \cdot)$ is a splitting function. In contrast, the **(ii) pairwise** approach can be viewed as a best-of-$n$ method, selecting a single best response from all candidates:

$$\hat{y} = f_{\text{pair}}(\mathcal{R}, \{y_i\}_{i=1}^n), \quad \mathcal{R} \sim r_\theta\left(x, \{y_i\}_{i=1}^n\right), \hat{y} \in \{y_i\}_{i=1}^n, \tag{11}$$

where $f_{\text{pair}}(\cdot, \cdot)$ is a selection function and $n = 2$ in most cases. Though the pairwise approach could be extended to $n > 2$, it could not be applied to single response scoring ($n = 1$).

**Representative Methods** Figure 2 illustrates how the three reward generation paradigms (scalar, semi-scalar, generative) can be combined with the two scoring patterns (pointwise, pairwise). Specifically, Bradley-Terry model (Kendall & Smith, 1940) ((a)+(i)) is trained with pairwise preference data and outputs scalar rewards pointwisely

$$\{S_i\}_{i=1}^n = f_{\text{point}}\left(\mathcal{R}, \{y_i\}_{i=1}^n\right) = \boldsymbol{S} \in \mathbb{R}^n. \tag{12}$$

PairRM (Jiang et al., 2023) ((a)+(ii)) compares a pair of responses with the sign of the scalar reward

$$\hat{y} = f_{\text{pair}}\left(\mathcal{R}, \{y_i\}_{i=1}^n\right) = y_{\lfloor \frac{1}{2}(3 - \text{sgn}(\boldsymbol{S})) \rfloor}, \quad n = 2, \mathcal{S} \in \mathbb{R}. \tag{13}$$

The scalar methods above could barely perform inference-time scaling due to the lack of diversity in reward generation. CLoud (Ankner et al., 2024) ((b)+(i)) generates scalar rewards for each response based on pre-generated critiques, similar to Equation 12. LLM-as-a-Judge (Zheng et al., 2023; Wang et al., 2025a) ((c)+(ii)) judges the preference order between paired responses textually,

$$\hat{y} = f_{\text{pair}}\left(\mathcal{R}, \{y_i\}_{i=1}^n\right) = y_{f_{\text{extract}}(\boldsymbol{C})}, \quad n = 2, \tag{14}$$

where $f_{\text{extract}}(\cdot)$ extracts the index of best response from language representations. However, this approach defaults to neglect ties of the paired responses. Following Zhang et al. (2025a), the generation probability of the token that indicates the preference order could be used as the scalar reward ((b)+(ii)): $\mathcal{S} = \text{TokenProb}(\hat{C}) = r_\theta(\hat{C}|x, \{y_i\}_{i=1}^n)$, where $\hat{C}$ is a pre-defined token related to the preference order. However, without additional constraints, GRMs are able to generate pointwise rewards for multiple responses within pure language representation ((c)+(i)):

$$\{S_i\}_{i=1}^n = f_{\text{point}}\left(\mathcal{R}, \{y_i\}_{i=1}^n\right) = f_{\text{extract}}(\boldsymbol{C}), \tag{15}$$

where $f_{\text{extract}}(\cdot)$ extracts the rewards assigned to each response from generation results. Usually, the rewards are discrete, and in this work we assign $S_i \in \mathbb{N}, 1 \leq S_i \leq 10$ by default. This approach promisingly allows both inference-time scalability and input flexibility.

## D.2 MODEL TRAINING

For the rule-based online RL, we use the standard GRPO setting (Shao et al., 2024). The overall objective is

$$\mathcal{J}_{\text{GRPO}}(\theta) = \mathbb{E}[q \sim P(Q), \{o_i\}_{i=1}^G \sim \pi_{\theta_{old}}(O|q)] \frac{1}{G} \sum_{i=1}^G \frac{1}{|o_i|} \sum_{t=1}^{|o_i|}$$

$$\left\{ \min\left[ \frac{\pi_\theta(o_{i,t}|q, o_{i,<t})}{\pi_{\theta_{old}}(o_{i,t}|q, o_{i,<t})} \hat{A}_{i,t}, \text{clip}\left( \frac{\pi_\theta(o_{i,t}|q, o_{i,<t})}{\pi_{\theta_{old}}(o_{i,t}|q, o_{i,<t})}, 1 - \epsilon, 1 + \epsilon \right) \hat{A}_{i,t} \right] - \beta \mathbb{D}_{KL}\left[\pi_\theta || \pi_{ref}\right] \right\}, \tag{16}$$

where $\hat{A}_{i,t} = \frac{\hat{r}_i - \text{mean}(\hat{\mathbf{r}})}{\text{std}(\hat{\mathbf{r}})}$, $G$ is the group size, $\beta$ is the coefficient of KL penalty, and $q = (x, \{y_i\}_{i=1}^n)$ with prompts. We performed grid search on hyper-parameter $\beta \in \{0.00, 0.01, 0.02, 0.08\}$ and found that $\beta = 0.08$ is the most stable configuration for SPCT-GRM-27B. And with too small KL coefficient, SPCT-GRM-27B tends to collapse on a few subsets in benchmarks, e.g., Chat in the Reward Bench benchmark and Harmlessness in the RMB benchmark, and shows biases towards some other domains. For smaller SPCT-GRM-16B, we use $\beta = 0.002$ because it is less vulnerable to the KL loss coefficient. We set $G = 4$ for a better trade-off between efficiency and performance.

The training set comprises of 1256K RFT data, including 1070K general instruction data and 186K rejective sampled data, and 237K RL data. General instruction data are from in-house datasets. Rejective sampled data and RL data are from the same RM datasets, containing the preference for single, paired, and multiple responses, constructed from internal data and open-source datasets, including the training sets from MATH (Hendrycks et al., 2021), UltraFeedback (Cui et al., 2024), OffsetBias (Park et al., 2024), Skywork-Reward-Preference-80K-v0.2 (Liu et al., 2024), and HelpSteer2-Preference (Wang et al., 2025b). Specifically, we re-tagged the preference label of a part of UltraFeedback due to its quality issues; we sampled and filtered trajectories on MATH by rule-based ground-truth matching, resulting in pairwise preference data; for rating single responses, we set the ground-truth reward to 1 for correct responses and 0 for incorrect ones, only incorporating verifiable

questions. For rejective sampling, we use DeepSeek-v2.5-0905 to generate the trajectories with principles and critiques. The sampling time $N_{\mathrm{RFT}}$ is set to 3. During hinted sampling on HelpSteer2, we add the preference strengths labeled in the original dataset as the hint. We also remove the samples that are viewed too easy for DeepSeek-V2-Lite-Chat, i.e. all generated rewards are correct for three times according to Equation 4, from the RL data.

The derivation of SPCT-GRM models and the meta RM is illustrated in Figure 5. All SPCT-GRM models are trained from the pretrained version of LLMs. For the training of the meta RM, we reuse the rejective sampled data from the RFT stage, and use SPCT-GRM-27B to perform rejective sampling with $N_{\mathrm{RFT}} = 3$, in order to avoid potential bias (Chow et al., 2025) in the meta RM guided voting. The learning rate is $1 \times 10^{-5}$ and the batch size is $512$ for the meta RM training. The training time of RFT and RL for SPCT-GRM-27B is depicted in Table 5, Gemma-2-27B based models are trained with 128 A100 GPUs on the Fire-Flyer platform (An et al., 2024). The learning rate is $5 \times 10^{-6}$ for the RFT stage and $4 \times 10^{-7}$ for the RL stage,

Table 5: Training times of RFT and RL stages for SPCT-GRM-27B in hours.

| Stage | Time (h) |
| --- | --- |
| RFT | 19.2 |
| Rule-Based RL | 15.6 |

and the batch size is $1024$ for the RFT stage and $512$ for the RL stage. Both stages are trained for 900 steps. Due to resource constraints, SPCT-GRM models larger than 27B does not undergo the rule-based RL and only trained with 50K rejective sampled data.

### D.3 BASELINE IMPLEMENTATION

For the baseline methods, we re-implement **LLM-as-a-Judge** (Zheng et al., 2023), **BT-RM-27B** (Kendall & Smith, 1940), **CLoud-Gemma-2-27B** (Ankner et al., 2024), and **PairRM-27B** (Jiang et al., 2023) based on Gemma-2-27B (Team, 2024) and with all compatible training data and settings as SPCT-GRM.

For **LLM-as-a-Judge**, we use exactly the same training configuration as SPCT-GRM-27B, including RFT with rejective sampled data from DeepSeek-v2.5-0905 and rule-based online RL. Due to its scoring pattern, only pairwise data could be used in the RL stage. For **CLoud-Gemma-2-27B**, we also generate pointwise critiques from DeepSeek-v2.5-0905 using the same prompt template. However, it is not feasible to perform rejective sampling, since no rewards could be extracted without a trained value head. We fine-tune Gemma-2-27B with the same general instruction data of SPCT-GRM-27B along with the sampled critique, resulting in a critique generation model. Specifically, we fine-tune another Gemma-2-27B model with a value head for reward generation, instead of training value heads post hoc on the critique model. The training of the value head of CLoud-Gemma-2-27B, **BT-RM-27B**, and **PairRM-27B** (Jiang et al., 2023) uses the same dataset from the RL stage of SPCT-GRM-27B, except for single response rating data.

## E EXPERIMENT DETAILS

### E.1 HYPER-PARAMETERS

For inference-time scaling results of SPCT-GRM-27B, SPCT-GRM-16B, LLM-as-a-Judge, and CLoud-Gemma-2-27B, the temperature is set to 0.5 for each model. And for other experiments, temperature is set to 0 for all models. Without specific description, $k_{\mathrm{meta}} = \frac{1}{2}k$ by default in the meta RM guided voting for SPCT-GRM-27B. For inference on DeepSeek-R1-0120, the temperature is set to 0.6. Please note that we let SPCT-GRM to output rewards in the same range for rating single responses in the ReaLMistake benchmark as other benchmarks.

### E.2 BENCHMARKS

We evaluate the performance of different methods on various RM benchmarks of different domains: (1) **Reward Bench** (Lambert et al., 2024), a common benchmark for RM evaluation, with semi-automatically collected chat (Li et al., 2023; Zheng et al., 2023; Zeng et al., 2024), reasoning (Lightman et al., 2024; Muennighoff et al., 2024), and safety (Röttger et al., 2024; Wang et al., 2024d) preference data, where two responses require to be ranked for each query; (2) **PPE** (Frick et al., 2025), a large-scale benchmark containing crowdsourced preference data and correctness data for

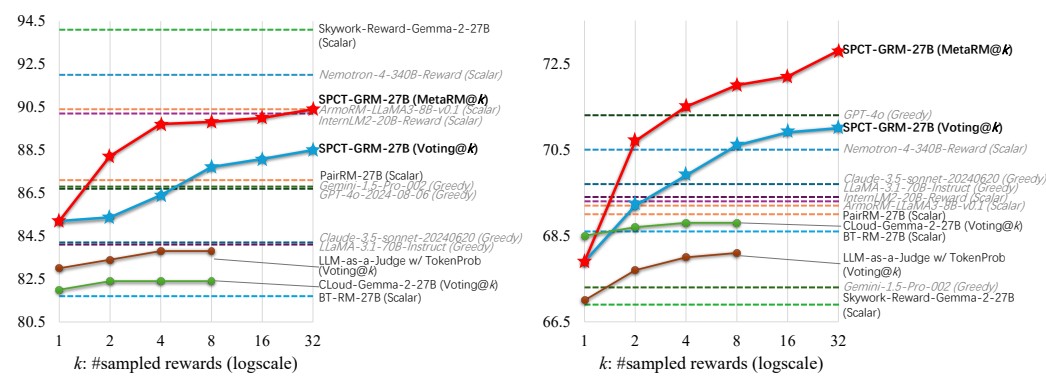

(a) Results on the Reward Bench benchmark. (b) Results on all tested reward modeling benchmarks.

Figure 6: Inference-time scaling performance with different RMs on different reward modeling benchmarks. Non-italic font indicates models based on Gemma-2-27B.

Table 6: Detailed results of inference-time scalability experiments (Table 3) of different methods and models on RM benchmarks. Underlined numbers indicate the best performance, **bold numbers** indicate the best performance among baseline and our methods, and *italicized font* denotes scalar or semi-scalar RMs. For meta RM guided voting (MetaRM), $k_{\mathrm{meta}} = \frac{1}{2}k$. Numbers in the parentheses is the performance change after inference-time scaling.

| Model | Reward Bench | PPE Preference | PPE Correctness | RMB | Overall |
|---|---|---|---|---|---|
| *Reported Results of Public Models* | | | | | |
| *Nemotron-4-340B-Reward* | *92.0* | *59.3* | *60.8* | *69.9* | *70.5* |
| GPT-4o | 86.7 | 67.1 | 57.6 | 73.8 | 71.3 |
| *Results of Inference-Time Scaling (Voting@1)* | | | | | |
| **LLM-as-a-Judge** | 83.0 | 63.4 | 57.4 | 64.3 | 67.0 |
| *CLoud-Gemma-2-27B* | *82.0* | *67.0* | *62.0* | *63.2* | *68.5* |
| **SPCT-GRM-27B-RFT (Ours)** | 84.0 | 62.2 | 59.4 | 65.8 | 67.8 |
| **SPCT-GRM-27B (Ours)** | 85.2 | 62.4 | 59.5 | 64.4 | 67.9 |
| *Results of Inference-Time Scaling (Voting@8)* | | | | | |
| **LLM-as-a-Judge** | 83.4 | 63.8 | 58.2 | 65.2 | 67.6 (+0.6) |
| **LLM-as-a-Judge w/ TokenProb** | 83.8 | 64.6 | 58.8 | 65.2 | 68.1 (+1.1) |
| *CLoud-Gemma-2-27B* | *82.4* | *67.3* | *62.4* | *63.2* | *68.8 (+0.3)* |
| **SPCT-GRM-27B-RFT (Ours)** | 85.3 | 64.5 | 59.7 | 67.7 | 69.3 (+1.5) |
| **SPCT-GRM-27B (Ours)** | 87.7 | 64.9 | 60.3 | 69.5 | 70.6 (+2.7) |
| **SPCT-GRM-27B (MetaRM) (Ours)** | 89.8 | 66.4 | 63.0 | 68.8 | 72.0 (+4.1) |
| *Results of Further Inference-Time Scaling (Voting@32)* | | | | | |
| **SPCT-GRM-27B (Ours)** | 88.5 | 65.3 | 60.4 | 69.7 | 71.0 (+3.1) |
| **SPCT-GRM-27B (MetaRM) (Ours)** | **90.4** | 67.2 | **63.2** | **70.3** | **72.8** (+4.9) |

verifiable tasks, and each query has two responses; (3) **RMB** (Zhou et al., 2025), a more comprehensive benchmark with various types of preference data, focusing on helpfulness and harmlessness, and each query has two responses or more response in pairwise and best-of-N (BoN) subsets, respectively; (4) **ReaLMistake** (Kamoi et al., 2024), a benchmark for diagnosing the error within single responses. Specifically, we do not include the prior sets (Bai et al., 2022a; Askell et al., 2021; Ethayarajh et al., 2022; Stiennon et al., 2020) of the Reward Bench benchmark in overall score calculations. For the reported results of public models, we use the scores released with each benchmark. Specifically, the version of gpt-4o is slightly different, as we reported results of gpt-4o-2024-08-06 for Reward Bench and PPE (the Correctness subset is reproduced with AlpacaEval prompt templates), and gpt-4o-2024-05-13 for RMB.

We use the standard evaluation metrics for each benchmark: accuracy of picking the best response from a set of responses in Reward Bench, PPE, and RMB, and ROC-AUC for ReaLMistake. The BoN subsets of the RMB benchmark contains multiple responses for each query, and each data point is correct only when the best response is identified. The default setting to evaluate models on RMB BoN subsets is to pairwise evaluate $(n-1)$ pairs, where each pair includes the best response and another different response, if there is totally $n$ responses. For baseline methods, we adopt this

Table 7: Detailed results of ablation studies (Table 4) for different components of the proposed SPCT. **Bold numbers** indicate the best performance.

| Model | Reward Bench | PPE Preference | PPE Correctness | RMB | Overall |
|---|---|---|---|---|---|
| *Results of Greedy Decoding* | | | | | |
| **SPCT-GRM-27B** | **86.0** | **64.7** | **59.8** | **69.0** | **69.9** |
| w/o Principle Generation | 82.0 | 62.8 | 58.2 | 67.1 | 67.5 |
| w/o Rejective Sampling | 84.0 | 63.2 | 59.4 | 68.0 | 68.7 |
| **SPCT-GRM-27B-RFT** | **84.5** | **64.1** | **59.6** | **67.0** | **68.8** |
| w/o Hinted Sampling (①) | 83.0 | 63.8 | 58.2 | 65.8 | 68.0 |
| w/o Non-Hinted Sampling (②) | 82.5 | 63.4 | 58.6 | 65.2 | 67.4 |
| w/o Rejective Sampling (①&②) | 81.5 | 61.8 | 57.8 | 63.1 | 66.1 |
| w/o General Instruction Data | 79.1 | 59.2 | 51.5 | 63.2 | 63.3 |
| *Results of Inference-Time Scaling (Voting@8)* | | | | | |
| **SPCT-GRM-27B** | **87.7** | **64.9** | **60.3** | **69.5** | **70.6** |
| w/o Principle Generation | 83.0 | 63.2 | 58.6 | 67.1 | 68.0 |
| *Results of Inference-Time Scaling (Voting@32)* | | | | | |
| **SPCT-GRM-27B** | 88.5 | 65.3 | 60.4 | 69.7 | 71.0 |
| **SPCT-GRM-27B** ($k_{\text{meta}} = 1$) | 88.5 | 67.1 | **65.2** | 65.2 | 71.5 |
| **SPCT-GRM-27B** ($k_{\text{meta}} = 8$) | 89.7 | **67.2** | 64.7 | 69.1 | 72.7 |
| **SPCT-GRM-27B** ($k_{\text{meta}} = 16$) | **90.4** | 67.2 | 63.2 | **70.3** | **72.8** |

Table 8: Detailed results of different methods on the Reward Bench benchmark. Underlined numbers indicate the best performance, **bold numbers** indicate the best performance among baseline and our methods, and *italicized font* denotes scalar or semi-scalar RMs. For meta RM guided voting (MetaRM), $k_{\text{meta}} = \frac{1}{2}k$.

| Method | Chat | Chat Hard | Safety | Reasoning | Prior Sets | Reward Bench |
|---|---|---|---|---|---|---|
| *Results of Other Models* | | | | | | |
| DeepSeek-R1-0120 | 97.1 | 73.7 | 73.3 | 95.6 | - | 84.9 |
| SPCT-GRM-16B | 90.8 | 74.3 | 84.7 | 81.8 | 62.5 | 82.9 |
| SPCT-GRM-230B | 96.5 | 72.5 | 87.8 | 84.3 | - | 85.3 |
| SPCT-GRM-671B | 95.8 | 82.9 | 88.3 | 86.6 | - | 88.4 |
| *Results of Greedy Decoding* | | | | | | |
| **LLM-as-a-Judge** | 96.7 | 69.3 | 83.5 | 84.3 | - | 83.4 |
| *BT-RM-27B* | 96.7 | 86.2 | 75.7 | 89.8 | 68.5 | *81.7* |
| *CLoud-Gemma-2-27B* | 96.7 | 69.3 | 83.5 | 84.3 | - | *82.0* |
| *PairRM-27B* | 95.5 | 86.8 | 52.3 | 92.0 | 67.6 | *87.1* |
| **SPCT-GRM-27B-RFT (Ours)** | 94.7 | 77.2 | 87.0 | 79.2 | 65.9 | 84.5 |
| **SPCT-GRM-27B (Ours)** | 94.1 | 78.3 | 88.0 | 83.8 | 66.7 | 86.0 |
| *Results of Inference-Time Scaling (Voting@8)* | | | | | | |
| **LLM-as-a-Judge** | 95.0 | 70.0 | 83.5 | 85.0 | - | 83.4 |
| **LLM-as-a-Judge w/ TokenProb** | 95.8 | 71.3 | 83.3 | 84.8 | - | 83.8 |
| *CLoud-Gemma-2-27B* | 96.7 | 85.8 | 56.2 | 91.0 | - | *82.4* |
| **SPCT-GRM-27B-RFT (Ours)** | 94.7 | 79.0 | 87.3 | 80.2 | - | 85.3 |
| **SPCT-GRM-27B (Ours)** | 95.3 | 80.9 | 89.3 | 85.4 | 66.8 | 87.7 |
| **SPCT-GRM-27B (MetaRM) (Ours)** | 95.5 | 85.7 | 88.5 | 89.5 | 69.4 | 89.8 |
| *Results of Further Inference-Time Scaling (Voting@32)* | | | | | | |
| **SPCT-GRM-27B (Ours)** | 95.5 | 81.8 | 90.0 | 86.9 | 68.1 | 88.5 |
| **SPCT-GRM-27B (MetaRM) (Ours)** | 95.3 | 85.7 | 89.5 | 91.0 | 69.4 | **90.4** |

approach for evaluation. And for our models (SPCT-GRM), we directly input all responses to the model and identify the best response with $\arg\max_i S_{i\,i=1}^{n}$, where $S_i$ is the predicted reward for $i$-th response, which is a more direct but harder way, and barely affects the performance. Please refer to Appendix F.1.1 for empirical analysis.

For DeepSeek-R1-0120, due to the large costs and latency of inference, we evenly down-sampled 300 data points from the Reward Bench benchmark, and test DeepSeek-R1-0120 on this subset. The result is illustrated in Figure 4(b).

### E.3 DETAILED RESULTS

We provide detailed results of Figure 1 in Figure 6, with performance of more public models for reference. We provide detailed results of Table 3 in Table 6, and detailed results of Table 4 in Table 7, with scores on each RM benchmark. Furthermore, we list detailed results for all tested methods on

Table 9: Detailed results of different methods on the PPE Correctness benchmark. Underlined numbers indicate the best performance, **bold numbers** indicate the best performance among baseline and our methods, and *italicized font* denotes scalar or semi-scalar RMs. For meta RM guided voting (MetaRM), $k_{\text{meta}} = \frac{1}{2}k$.

| Method | MMLU-Pro | MATH | GPQA | MBPP-Plus | IFEval | PPE Correctness |
|---|---|---|---|---|---|---|
| *Results of Greedy Decoding* | | | | | | |
| **LLM-as-a-Judge** | 66.0 | 68.0 | 52.8 | 50.2 | 56.8 | 58.8 |
| *BT-RM-27B* | *68.8* | *73.2* | *56.8* | *68.8* | *66.0* | ***66.7*** |
| *CLoud-Gemma-2-27B* | *68.7* | *68.8* | *53.5* | *59.0* | *62.0* | *62.4* |
| *PairRM-27B* | *68.3* | *74.7* | *55.0* | *63.1* | *62.9* | *64.8* |
| **SPCT-GRM-27B-RFT (Ours)** | 64.8 | 68.7 | 55.5 | 49.0 | 60.2 | 59.6 |
| **SPCT-GRM-27B (Ours)** | 64.8 | 68.8 | 55.6 | 50.1 | 59.8 | 59.8 |
| w/ Reference | 98.2 | 97.5 | 99.8 | 86.6 | 75.9 | 91.6 |
| *Results of Inference-Time Scaling (Voting@8)* | | | | | | |
| **LLM-as-a-Judge** | 66.2 | 66.4 | 51.9 | 49.9 | 56.8 | 58.2 |
| **LLM-as-a-Judge** w/ TokenProb | 66.4 | 68.1 | 53.0 | 49.5 | 57.0 | 58.8 |
| *CLoud-Gemma-2-27B* | *68.7* | *68.9* | *53.5* | *59.0* | *62.0* | *62.4* |
| **SPCT-GRM-27B-RFT (Ours)** | 64.8 | 68.7 | 55.5 | 49.5 | 60.2 | 59.7 |
| **SPCT-GRM-27B (Ours)** | 65.7 | 68.7 | 55.5 | 50.0 | 61.6 | 60.3 |
| **SPCT-GRM-27B (MetaRM) (Ours)** | 68.0 | 68.7 | 57.3 | 51.3 | 69.9 | 63.0 |
| *Results of Further Inference-Time Scaling (Voting@32)* | | | | | | |
| **SPCT-GRM-27B (Ours)** | 65.5 | 69.4 | 56.0 | 49.9 | 61.0 | 60.4 |
| **SPCT-GRM-27B (MetaRM) (Ours)** | 68.1 | 70.0 | 56.9 | 50.8 | 70.4 | **63.2** |

Table 10: Detailed results of different methods on the RMB benchmark.

| Method | Helpfulness BoN | Helpfulness Pairwise | Harmlessness BoN | Harmlessness Pairwise | RMB |
|---|---|---|---|---|---|
| *Results of Greedy Decoding* | | | | | |
| **LLM-as-a-Judge** | 55.8 | 78.5 | 50.8 | 73.9 | 64.8 |
| *BT-RM-27B* | *64.0* | *83.0* | *33.6* | *51.0* | *57.9* |
| *CLoud-Gemma-2-27B* | *64.7* | *81.1* | *41.7* | *66.1* | *63.4* |
| *PairRM-27B* | *59.9* | *83.3* | *34.1* | *55.5* | *58.2* |
| **SPCT-GRM-27B-RFT (Ours)** | 58.4 | 79.3 | 54.2 | 76.0 | 67.0 |
| **SPCT-GRM-27B (Ours)** | 62.3 | 80.5 | 57.0 | 76.1 | 69.0 |
| *Results of Inference-Time Scaling (Voting@8)* | | | | | |
| **LLM-as-a-Judge** | 56.0 | 78.5 | 52.5 | 73.8 | 65.2 |
| **LLM-as-a-Judge** w/ TokenProb | 56.0 | 78.5 | 52.5 | 73.8 | 65.2 |
| *CLoud-Gemma-2-27B* | *63.8* | *82.1* | *40.9* | *66.1* | *63.2* |
| **SPCT-GRM-27B-RFT (Ours)** | 59.2 | 80.1 | 54.8 | 76.5 | 67.7 |
| **SPCT-GRM-27B (Ours)** | 63.9 | 79.5 | 57.6 | 77.1 | 69.5 |
| **SPCT-GRM-27B (MetaRM) (Ours)** | 63.4 | 80.5 | 56.8 | 74.6 | 68.8 |
| *Results of Further Inference-Time Scaling (Voting@32)* | | | | | |
| **SPCT-GRM-27B (Ours)** | 63.9 | 79.8 | 58.0 | 77.0 | 69.7 |
| **SPCT-GRM-27B (MetaRM) (Ours)** | 64.2 | 81.6 | 58.0 | 77.4 | **70.3** |

each RM benchmarks, with the Reward Bench benchmark in Table 8, the PPE Correctness benchmark in Table 9, and the RMB benchmark in Table 10. We found that DeepSeek-R1 achieves the highest result in the Reasoning subset of the Reward Bench benchmark, indicating that long-horizon reasoning could boost GRMs in reasoning extensive scenarios.

# F ADDITIONAL EXPERIMENTS

## F.1 INPUT FLEXIBILITY OF THE POINTWISE GRM APPROACH

In Section 2.1, we demonstrate the input flexibility of the pointwise GRM approach theoretically. In this section, we provide empirical evidence on various input types to support it.

### F.1.1 GENERATING REWARDS FOR MANY RESPONSES

In Table 11, we show the experimental results of SPCT-GRM-27B on the BoN subsets of the RMB benchmark, where each query has multiple responses. If there is at total $n, (n > 2)$ responses for a query, the pair input setting is to evaluate $(n − 1)$ pairs comprise of the best response and the other responses, and only when the best response is correctly identified from all $(n − 1)$ pairs, the data point is considered as correct. It is also the default setting for the original benchmark. We compare

Table 11: Experiments of response input types on the RMB BoN benchmarks.

| Method | Helpfulness | Harmlessness |
|--------|-------------|--------------|
| **SPCT-GRM-27B** | | |
| w/ Pair Input | 62.1 | 57.5 |
| w/ List Input | 62.3 | 57.0 |
| $|\Delta|$ | 0.2 | 0.5 |

Table 12: Experiments on reference-based RM on the PPE correctness benchmark.

| Method | Overall |
|--------|---------|
| **SPCT-GRM-27B** | 59.8 |
| w/ Voting@32 | 60.4 |
| w/ Meta RM ($k_{\text{meta}} = 8$) | 64.7 |
| w/ Reference | 91.6 |

Table 13: Experimental results (ROC-AUC (%)) on the ReaLMistake benchmark.

| Model | Overall |
|-------|---------|
| DeepSeek-V2.5-0905 | 69.4 |
| GPT-4o-2024-08-06 | 74.3 |
| DeepSeek-V2-Lite-Chat | 61.9 |
| **SPCT-GRM-16B (Ours)** | 64.9 |
| Gemma-2-27B-it | 65.8 |
| **BT-RM-27B** | 69.3 |
| **SPCT-GRM-27B (Ours)** | 72.2 |
| **SPCT-GRM-27B (Voting@8) (Ours)** | 74.4 |

the performance of SPCT-GRM-27B with pair input and list input, where the list input setting is to identify the best response with inputting all $n$ responses. The result shows that SPCT-GRM-27B is barely affected by the input types, and the performance difference is less than 1% on both helpfulness and harmlessness subsets. This indicates that **the pointwise GRM is flexible to input many responses, and the performance is not sensitive to the input types**.

### F.1.2 GENERATING REWARDS FOR SINGLE RESPONSES

In Table 13, we show the experimental results of SPCT-GRM in 16B and 27B on the ReaLMistake benchmark, where each query has only one response. We compare with public models, e.g., DeepSeek-V2.5-0905, GPT-4o-2024-08-06, DeepSeek-V2-Lite, and Gemma-2-27B-it, and BT-RM-27B. The result shows that SPCT-GRM achieves the best performance among models with the same size, and comparable performance with the best public models with inference-time scaling. This indicates that **the pointwise GRM could effectively rate single responses**.

### F.1.3 GENERATING REWARDS WITH REFERENCE

In Section 5.2, we show that scalar and semi-scalar RMs could have significant domain biases, and generally perform better on verifiable questions. To alleviate this issue, we test SPCT-GRM-27B to generate rewards for these tasks with reference, where the reference is the ground truth for each query. The results are shown in Table 12. We find that SPCT-GRM-27B could achieve a more than 90% accuracy with reference provided. This indicates that **the pointwise GRM could effectively judge responses with reference, mitigating performance on verifiable tasks**.

### F.2 TRANSFERABILITY OF GENERATED PRINCIPLES

We extend the preliminary experiment in Section 2.2 with SPCT-GRM-27B generated principles. We test GPT-4o-2024-08-06 and SPCT-GRM-27B with the filtered principles exactly the same as Table 1, and aforementioned SPCT-GRM-27B generated ones. The results are shown in Table 14. We find that the principles generated by SPCT-GRM-27B could be transferred to other models, and are even sightly better than manually filtered principles from GPT-4o. This indicates that **the principles generated by SPCT-GRM-27B are robust and transferable to other models**.

Table 14: Experiments of the transferability of principles generated by different models.

| Method | Chat Hard | IFEval |
|--------|-----------|--------|
| **GPT-4o-2024-08-06** | 76.1 | 56.0 |
| +Self-Gen. Principles | 75.9 | 55.6 |
| +Filtered Principles | 77.8 | 57.5 |
| +SPCT-GRM-27B-Gen. Principles | 78.1 | 58.3 |
| **SPCT-GRM-27B** | 78.3 | 59.8 |
| +Filtered Principles | 77.0 | 58.5 |

### F.3 GENERALIZATION BEYOND TRAINING DATA

Table 15: Results of training data generalization experiments on the Reward Bench benchmark. **Bold numbers** indicate the best performance.

| Model | Chat | Chat Hard | Safety | Reasoning | Reward Bench |
|---|---|---|---|---|---|
| *Results of Greedy Decoding* | | | | | |
| **SPCT-GRM-27B** | **94.1** | **78.3** | **88.0** | **83.8** | **86.0** |
| w/o MATH RM Data | 96.1 | 70.4 | 85.3 | 82.5 | 83.0 |
| **SPCT-GRM-16B** | **90.8** | **74.3** | **84.7** | **81.8** | **82.9** |
| w/o MATH RM Data | 95.0 | 63.4 | 76.9 | 74.3 | 77.4 |

We conduct ablation study on the generalization of training data for SPCT-GRM-27B. We remove the all data from MATH training set, and re-implement the training recipe. Results on the Reward Bench benchmark are shown in Table 15. We found that merely adding math related preference data could also boost generalist RM performance on various domains, especially on the Chat Hard subset. The result reveals that SPCT-GRM-27B could generalize to domains beyond the coverage of training data.

### F.4 RESPONSE LENGTH ANALYSIS FOR RULE-BASED RL

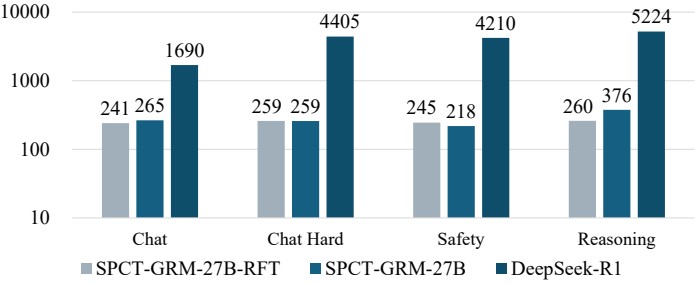

Figure 7: The changes of response lengths (#tokens) of SPCT-GRM-27B before and after rule-based online RL on the Reward Bench benchmark, compared with DeepSeek-R1-0120.

We calculate the response lengths of SPCT-GRM-27B before and after rule-based online RL on each subset of the Reward Bench benchmark in Figure 7. The token count of SPCT-GRM-27B is calculated based on the tokenizer of Gemma-2-27B, while the result of DeepSeek-R1-0120 uses its corresponding tokenizer. We found that the response length for the Chat subset barely increases in RL, and the response length for the Safety subset even drops slightly. The largest increase of response lengths occurs in the Reasoning subset, where the performance of SPCT-GRM-27B also improves greatly compared to SPCT-GRM-27B-RFT, according to Table 8. This might indicate that SPCT-GRM-27B learns to adaptively use more inference compute on reasoning extensive tasks, and the compute could be saved for some other domains, such as safety, after the model learns to generate principles accurately. However, DeepSeek-R1-0120 uses way more tokens and achieves lower results, except for Reasoning, which shows that long-horizon reasoning also helps RM tasks regarding to extensive reasoning.

## G QUALITATIVE ANALYSIS

### G.1 CASE STUDY

We provide a case study on SPCT-GRM-27B in Table 16, 17 and 18. The first case shows that BT-RM-27B as a scalar RM could be hacked or biased under specific circumstances, and SPCT-GRM-27B generates textual principles and critiques, showing better robustness. The second case shows

the scalable behaviors of SPCT-GRM-27B, generating accurate rewards after voting on multiple samples. The according meta RM scores also show the effectiveness of the meta RM in guiding the voting process. The third case shows the potential failure of SPCT-GRM-27B which is caused by the inability of the model to accurately judge responses following some principles, e.g., assessing real-time data, and the weights of each principle might not be balanced. The critique processes including weight attribution and score collection are largely emerged from RFT and online RL training, which can largely vary among different samplings and test cases.

Table 16: A case of SPCT-GRM-27B and BT-RM-27B, where SPCT-GRM-27B shows better robustness in distinguishing slight differences of responses.

| **Query** | Please help me review for my behavioral neuroscience exam by giving me the core information that is essential to this course. Please give an expansive review and include explanations in analogy or metaphors |
|---|---|

| **Response 1** | Behavioral neuroscience, also known as biological psychology, explores the relationship between the brain and behavior. It examines how the nervous system influences cognitive processes, emotions, and motor functions. Here's a comprehensive review with analogies and metaphors:\n |
| --- | --- |
| | 1. **Neuroanatomy**: Imagine the brain as a complex city with different neighborhoods (lobes) and roads (neuronal pathways). The cerebral cortex is the outer layer, like the skin of the city, responsible for higher functions like thinking and perception. The cerebellum is like a choreographer, ensuring smooth and coordinated movements. The brainstem acts as the city's backbone, connecting the brain with the spinal cord, and controlling basic life functions. |
| | 2. **Neurons**: These are the building blocks of the nervous system. Think of them as the city's messengers, transmitting information via electrical and chemical signals. The dendrites receive messages, the cell body processes them, and the axon sends the signal to other neurons at the synapse. |
| | 3. **Synapses and neurotransmitters**: The synapse is like a tiny bridge between neurons. Neurotransmitters are the messengers that cross this bridge, transmitting signals from one neuron to another. Analogous to different languages, there are various neurotransmitters (e.g., dopamine, serotonin, glutamate) that influence distinct aspects of behavior and mood. |
| | 4. **Sensory Systems**: These are the city's surveillance cameras, detecting external stimuli. For example, the visual system (retina) captures light, and the auditory system (cochlea) picks up sound waves. These systems transmit information to the brain for processing. |
| | 5. **Motor Systems**: The motor systems act like the city's transportation network, sending signals from the brain to muscles, enabling movement. The somatic motor system controls voluntary movements, while the autonomic nervous system is like an unseen hand managing involuntary functions like heart rate and digestion. |
| | 6. ** Plasticity**: Brain plasticity is like the city's ability to adapt and change over time. It refers to the brain's capacity to reorganize itself by forming new neural connections in response to experiences, learning, or injury. |
| | 7. **Memory**: Memory can be thought of as a vast library where information is stored and retrieved. There are three main types: sensory memory (like a quick glance at a book), short-term memory (reading a paragraph), and long-term memory (remembering the entire book). |
| | 8. **Learning**: Learning is the process of acquiring new knowledge, skills, or behaviors. It's like adding new books to the library or updating old information. There are different types, such as classical conditioning (learning through association, like Pavlov's dogs) and operant conditioning (learning through consequences, like reinforcements and punishments). |
| | 9. **Emotions and Motivation**: Emotions are like the city's weather, influencing how we perceive and respond to events. Different brain regions, like the amygdala (the emotional storm center), contribute to emotional experiences. Motivation can be seen as the city's energy, driving us to act in pursuit of goals, with systems like the mesolimbic dopamine pathway playing a key role. |
| | 10. **Hormones and the Endocrine System**: These are the city's internal postal service, sending chemical messages throughout the body. Hormones, like cortisol and oxytocin, can influence behavior and mood by interacting with the brain and nervous system.\n |
| | Understanding these core concepts will help you navigate the complex world of behavioral neuroscience and prepare for your exam. Remember that this field is constantly evolving, so stay curious and engaged with the latest research. |

| | |
|---|---|
| **Response 2** | Certainly! Behavioral neuroscience is a fascinating field that explores the intricate relationship between the brain and behavior. It's like unraveling the mysteries of a complex machine, where the brain is the central processing unit, and behavior is the output. Let's dive into the core concepts that form the foundation of this course.\n
| | 1. Neuroanatomy: Understanding the Brain's Architecture\n The brain is a remarkable structure, and understanding its anatomy is crucial. Think of it as a vast city, with different regions serving as specialized districts. The cerebral cortex is the bustling metropolis, where higher cognitive functions like decision-making and problem-solving take place. The limbic system, on the other hand, is the emotional heart, regulating feelings and memories. And deep within, the brainstem acts as the city's control center, managing essential functions like breathing and heartbeat.\n
| | 2. Neurophysiology: The Language of the Brain\n Just as a city communicates through intricate networks of roads and signals, the brain uses electrical and chemical signals to transmit information. Neurons, the brain's messengers, fire action potentials like lightning bolts, carrying vital information. Neurotransmitters, on the other hand, are the chemical couriers, facilitating communication between neurons at synapses, akin to the handoff of a letter from one postal worker to another.\n
| | 3. Sensory and Motor Systems: The Brain's Gateways\n The brain is not an isolated entity; it constantly interacts with the external world. The sensory systems act as the brain's eyes, ears, and touch receptors, gathering information from the environment. Imagine them as surveillance cameras and sensors, feeding data into the brain's processing centers. Conversely, the motor systems are the brain's actuators, executing commands and translating neural signals into physical movements, like a robotic arm carrying out instructions.\n
| | 4. Learning and Memory: The Brain's Database\n Just like a computer stores and retrieves data, the brain has its own mechanisms for learning and memory. Long-term potentiation (LTP) is like upgrading the bandwidth between neurons, strengthening connections and facilitating information storage. Consolidation is akin to backing up data, transferring memories from short-term to long-term storage. And retrieval is like accessing a file, recalling stored information when needed.\n
| | 5. Motivation and Emotion: The Brain's Driving Forces\n Behavior is not just a product of cognitive processes; it's also influenced by motivation and emotion. The reward system, centered around structures like the nucleus accumbens, acts as the brain's carrot-and-stick, reinforcing desirable behaviors with pleasurable sensations. The amygdala, on the other hand, is the brain's security guard, processing emotional cues and triggering appropriate responses, like the fight-or-flight reflex.\n
| | 6. Neuroplasticity: The Brain's Adaptability\n The brain is not a static entity; it's a dynamic and ever-changing landscape. Neuroplasticity is the brain's ability to reorganize and rewire itself in response to experience and environmental changes. It's like a city constantly undergoing renovations, adapting to new demands and challenges. This plasticity allows the brain to recover from injuries, acquire new skills, and adapt to changing circumstances.\n
| | 7. Neurological Disorders: When the Brain Malfunctions\n Just as a city can experience breakdowns and malfunctions, the brain is susceptible to various disorders and dysfunctions. Neurodegenerative diseases like Alzheimer's and Parkinson's are akin to the gradual deterioration of a city's infrastructure, leading to cognitive and motor impairments. Psychiatric disorders, such as depression and anxiety, can be viewed as imbalances in the brain's chemical signaling system, disrupting emotional regulation and behavior.\n
| | By understanding these core concepts and embracing the analogies and metaphors, you'll gain a deeper appreciation for the intricate workings of the brain and its profound influence on behavior. Remember, the brain is a complex and fascinating realm, and behavioral neuroscience is the key to unlocking its secrets. |
| **BT-RM-27B Result** | Response 1: 0.4665; Response 2: 0.3209. |

| SPCT-GRM-27B Result | Specific Criteria:
1. **Depth of Analogies and Metaphors** (Weight: 30%): The extent to which the analogies and metaphors used are deep, creative, and effectively convey complex concepts.
2. **Comprehensiveness of Core Information** (Weight: 30%): The thoroughness with which the core information essential to behavioral neuroscience is covered.
3. **Clarity and Organization** (Weight: 20%): The clarity of explanations and the logical organization of the information.
4. **Engagement and Appeal** (Weight: 20%): The level of engagement and appeal the response has, making the information interesting and memorable.

Analysis:
- **Response 1**: This response excels in comprehensiveness and clarity, covering all major aspects of behavioral neuroscience with detailed and well-organized explanations. The analogies and metaphors are well-crafted and effectively enhance understanding. However, some metaphors could be more creative to enhance engagement further.
- **Response 2**: This response is also comprehensive and clear, but it stands out more in the depth and creativity of its analogies and metaphors. The organization is logical, and the explanations are clear. The engagement level is high, making the information appealing and memorable.

Scores: \boxed{8, 9} |
|---|---|
| **Ground Truth** | *Response 2 is better than Response 1.* |

Table 17: A case of SPCT-GRM-27B in the inference-time scaling setting. The input order of responses is reversed for SPCT-GRM-27B when generating result 2 and result 3.

| **Query** | Write a JavaScript function 'const tri = (n)' to solve the following problem:
Everyone knows Fibonacci sequence, it was studied deeply by mathematicians in
the last couple centuries. However, what people don't know is Tribonacci sequence.
Tribonacci sequence is defined by the recurrence:
tri(1) = 3
tri(n) = 1 + n / 2, if n is even.
tri(n) = tri(n - 1) + tri(n - 2) + tri(n + 1), if n is odd.
For example:
tri(2) = 1 + (2 / 2) = 2
tri(4) = 3
tri(3) = tri(2) + tri(1) + tri(4)
= 2 + 3 + 3 = 8
You are given a non-negative integer number n, you have to a return a list of the first n + 1 numbers of the Tribonacci sequence.
Examples:
tri(3) = [1, 3, 2, 8] |
|---|---|

| | |
|---|---|
| **Response 1** | ```
if (n == 0) { return [1] }
if (n == 1) { return [1, 3] }
let p = [1, 3]
for (let i = 2; i <= n; i++) {
    if (i % 2 == 0) {
        p.push(1 + i / 2)
    }
    else {
        p.push(p[i - 2] + p[i - 1] + 1 + i + (i + 1) / 2)
    }
}
return p
}
``` |
| **Response 2** | ```
if (n == 0) { return [1] }
if (n == 1) { return [1, 3] }
let p = [1, 3]
for (let i = 2; i <= n; i++) {
    if (i % 2 == 0) {
        p.push(1 + i / 2)
    }
    else {
        p.push(p[i - 2] + p[i - 1] + 1 + (i + 1) / 2)
    }
}
return p
}
``` |

| SPCT-GRM-27B Result 1 | ### Output |
|---|---|
| | **Specific Criteria:** |
| | 1. **Correctness of Sequence Calculation (Weight: 40%)**: The response must correctly implement the Tribonacci sequence as defined, including handling both even and odd cases accurately. |
| | 2. **Initialization and Base Cases (Weight: 20%)**: The response should correctly initialize the sequence and handle base cases (n = 0 and n = 1) appropriately. |
| | 3. **Loop and Conditional Logic (Weight: 20%)**: The response should use appropriate loop and conditional logic to generate the sequence. |
| | 4. **Output Format (Weight: 20%)**: The response should return the sequence in the correct format as specified. |
| | |
| | **Analysis:** |
| | - **Response 1**: |
| | - **Correctness of Sequence Calculation**: The logic for even and odd cases is mostly correct, but there is a mistake in the calculation for odd numbers where it incorrectly adds '1 + i + (i + 1) / 2'. |
| | - **Initialization and Base Cases**: Correctly initializes and handles base cases. |
| | - **Loop and Conditional Logic**: Uses appropriate loop and conditional logic. |
| | - **Output Format**: Returns the sequence in the correct format. |
| | |
| | - **Response 2**: |
| | - **Correctness of Sequence Calculation**: Similar to Response 1, the logic for even and odd cases is mostly correct, but there is a mistake in the calculation for odd numbers where it incorrectly adds '1 + (i + 1) / 2'. |
| | - **Initialization and Base Cases**: Correctly initializes and handles base cases. |
| | - **Loop and Conditional Logic**: Uses appropriate loop and conditional logic. |
| | - **Output Format**: Returns the sequence in the correct format. |
| | |
| | **Scores:** |
| | - **Response 1**: |
| | - Instruction Adherence: 8 (Partially Adhered)\n - Usefulness: 7 (Useful but Incomplete)\n - Level of Detail: 8 (Detailed but Slightly Lacking)\n - Relevance: 9 (Highly Relevant)\n - **Overall Score**: 8\n |
| | - **Response 2**: |
| | - Instruction Adherence: 8 (Partially Adhered)\n - Usefulness: 7 (Useful but Incomplete)\n - Level of Detail: 8 (Detailed but Slightly Lacking)\n - Relevance: 9 (Highly Relevant)\n - **Overall Score**: 8 |
| | |
| | **Scores: \boxed{8, 8}** |
| **Meta RM Score 1** | -15.7781 |

| | |
|---|---|
| **SPCT-GRM-27B Result 2** | ### Output

Specific Criteria:
1. **Correctness of Tribonacci Sequence Calculation** (Weight: 40%): The response must correctly implement the Tribonacci sequence as defined in the problem.
2. **Code Structure and Readability** (Weight: 30%): The code should be well-structured, readable, and follow best practices.
3. **Edge Case Handling** (Weight: 20%): The response should correctly handle edge cases such as 'n = 0' and 'n = 1'.
4. **Efficiency** (Weight: 10%): The code should efficiently compute the sequence without unnecessary computations.

Analysis:
- **Response 1**:
- **Correctness of Tribonacci Sequence Calculation**: The response correctly implements the Tribonacci sequence for both even and odd indices.
- **Code Structure and Readability**: The code is structured well and is readable. It uses a loop to generate the sequence and correctly handles the conditions for even and odd indices.
- **Edge Case Handling**: The response correctly handles the edge cases for 'n = 0' and 'n = 1'.
- **Efficiency**: The code efficiently computes the sequence with a single loop.

- **Response 2**:
- **Correctness of Tribonacci Sequence Calculation**: The response incorrectly computes the sequence for odd indices by adding an extra '1 + i + (i + 1) / 2', which deviates from the defined Tribonacci sequence.
- **Code Structure and Readability**: The code is structured similarly to Response 1 but includes an incorrect computation.
- **Edge Case Handling**: The response correctly handles the edge cases for 'n = 0' and 'n = 1'.
- **Efficiency**: The code is efficient but the incorrect computation affects its usefulness.

Scores:
\boxed{9, 5} |
| **Meta RM Score 2** | 1.3126 |

| | |
|---|---|
| **SPCT-GRM-27B Result 3** | ### Specific Criteria:
1. **Correctness of Tribonacci Sequence Calculation (Weight: 40%)**:
- The response must correctly implement the Tribonacci sequence as defined, including handling both even and odd cases accurately.
2. **Code Structure and Readability (Weight: 30%)**:
- The code should be well-structured, readable, and follow best practices in JavaScript.
3. **Edge Case Handling (Weight: 20%)**:
- The response should correctly handle edge cases such as 'n = 0' and 'n = 1'.
4. **Efficiency (Weight: 10%)**:
- The code should be efficient in terms of time and space complexity, avoiding unnecessary computations.

### Analysis:
- **Response 1**:
- **Correctness of Tribonacci Sequence Calculation**: The response correctly implements the Tribonacci sequence for both even and odd cases. It accurately follows the given recurrence relations.
- **Code Structure and Readability**: The code is structured well and is readable. It uses a loop to generate the sequence and handles the initial conditions correctly.
- **Edge Case Handling**: The response correctly handles the edge cases 'n = 0' and 'n = 1'.
- **Efficiency**: The code is efficient, with a time complexity of O(n) and space complexity of O(n).

- **Response 2**:
- **Correctness of Tribonacci Sequence Calculation**: The response has a logical error in the calculation for odd 'n'. It incorrectly adds '1 + i + (i + 1) / 2' instead of 'p[i - 2] + p[i - 1] + 1 + (i + 1) / 2'.
- **Code Structure and Readability**: The code structure is similar to Response 1, but the logical error makes it less readable.
- **Edge Case Handling**: The response correctly handles the edge cases 'n = 0' and 'n = 1'.
- **Efficiency**: The code is efficient in terms of time and space complexity, but the logical error impacts its correctness.

### Scores:
- **Response 1**:
- Instruction Adherence: 10\n - Usefulness: 10\n - Level of Detail: 9\n - Relevance: 10\n - **Overall Score**: 10\n
- **Response 2**:
- Instruction Adherence: 8\n - Usefulness: 6\n - Level of Detail: 8\n - Relevance: 8\n - **Overall Score**: 7

### Output:
\boxed{10, 7} |
| **Meta RM Score 3** | 1.6739 |
| **Voted Result** | Response 1: 20; Response 2: 27. |
| **Ground Truth** | *Response 2 is better than Response 1.* |

Table 18: A failure case of SPCT-GRM-27B due to the incapability of distinguish responses according to the principles, such as real-time accuracy examination.

| Query | User: Make a research and find the best coins opportunities and relatively obscure coins that have the potential for significant growth for a short-term & mid-term from coingecko and it must be listed in centralized exchange. Remove all coins with price higher than 1$. By implementing fundamental analysis to assessing the intrinsic value, news and crypto social sentiment and voulume of discussion analysis, On-Chain analysis, time series analysis and technical analysis, You will decide which cryptocurrencies is the best to invest. Then show me the best 5 coins that met the all of the conditions. Then show me the real time price of each coin. Then calculate the weekly support and resistance. Then show the best price to buy, for each coin. Then show me the target price. Then insert all of the above in a table and show it to me. |
|---|---|
| | Assistant: Based on the given criteria, here are the top 5 relatively obscure coins with potential for significant growth for short-term and mid-term investment. Please note that these recommendations should not be considered as financial advice, and are subject to change. |
| | \| Coin \| Real-time Price \| Weekly Support \| Weekly Resistance \| Best Price to Buy \| Target Price \|
\|——\|—————\|—————-\|—————\|—————-\|————\|
\| XTZ (Tezos) \| $0.70 \| $0.65 \| $0.80 \| $0.68 \| $1.2 \|
\| REN (Ren) \| $0.40 \| $0.35 \| $0.48 \| $0.38 \| $0.72 \|
\| LRC (Loopring) \| $0.08 \| $0.07 \| $0.10 \| $0.075 \| $0.14 \|
\| OCEAN (Ocean Protocol) \| $0.32 \| $0.28 \| $0.38 \| $0.305 \| $0.52 \|
\| ZRX (0x) \| $0.96 \| $0.89 \| $1.10 \| $0.94 \| $1.5 \| |
| | Please note that the real-time prices mentioned above will change according to the current market rate. It is vital to conduct your due diligence and re-evaluate the investments regularly. The target prices mentioned are speculative estimates based on assessed potential growth and technical analysis. Always exercise caution and consult a financial advisor before making any investment decisions. |
| | User: The real time prices aren't correct. |
| Response 1 | I apologize for the misunderstanding, as I do not have real-time data access to provide current cryptocurrency prices or market conditions. However, I can guide you on how to find this information and analyze it.\n
To identify potential investment opportunities in cryptocurrencies priced under $1, you can follow these steps:\n
1. **Use CoinGecko or CoinMarketCap**: These platforms allow you to filter cryptocurrencies by price, market cap, and trading volume. You can set the price filter to under $1.\n
2. **Fundamental Analysis**: Look for coins with strong fundamentals, such as:\n - Active development teams\n - Real-world use cases\n - Partnerships and collaborations\n - Community support and engagement\n
3. **News and Sentiment Analysis**: Use tools like Google Trends, Twitter, and Reddit to gauge the sentiment around specific coins. Positive news can drive short-term price increases.\n
4. **On-Chain Analysis**: Examine metrics like transaction volume, active addresses, and network growth to assess the health of the blockchain.\n
5. **Technical Analysis**: Use charting tools to identify support and resistance levels, trends, and potential entry and exit points.\n
6. **Volume of Discussion**: Monitor forums and social media platforms to see which coins are being discussed frequently, as this can indicate growing interest.\n
Once you gather this information, you can create a table similar to the one I provided earlier, but with accurate real-time data.\n
If you need help with specific analysis techniques or tools, feel free to ask! |

| | |
|---|---|
| **Response 2** | I apologize for any confusion in the real-time prices mentioned in the previous response. It is quite challenging to ensure real-time accuracy due to the rapidly changing nature of cryptocurrency prices. Here are the updated real-time prices at the time of this response.\n
\| Coin \| Real-time Price \|
\|———\|—————\|
\| XTZ (Tezos) \| \$0.63 \|
\| REN (Ren) \| \$0.36 \|
\| LRC (Loopring) \| \$0.08 \|
\| OCEAN (Ocean Protocol) \| \$0.29 \|
\| ZRX (0x) \| \$0.93 \|\n
To obtain accurate real-time prices, I recommend using reliable cryptocurrency tracking websites such as CoinGecko, CoinMarketCap, or Binance. Please note that the table containing other information provided earlier stands, as it is not dependent on real-time prices.\n
Keep in mind that investing in cryptocurrencies always carries risk, and it's essential to do your own research, consider your investment strategy, and exercise caution before making investment decisions. Always consult a financial advisor for professional guidance. |
| **SPCT-GRM-27B Result** | Specific Criteria:
- Real-time Price Accuracy (Weight: 30%): The response must provide accurate real-time prices for the listed coins.
- Guidance on Analysis Techniques (Weight: 20%): The response should offer clear guidance on how to conduct the required analyses.
- Table Update (Weight: 15%): The response should update the table with the correct real-time prices.
- Risk Disclosure (Weight: 10%): The response should include a clear disclaimer about the risks of investing in cryptocurrencies.
- Resource Recommendations (Weight: 10%): The response should recommend reliable resources for obtaining real-time data.
- Clarity and Structure (Weight: 15%): The response should be clear, well-structured, and easy to follow.

Analysis:
- **Response 1**: This response acknowledges the lack of real-time data access and provides a comprehensive guide on how to find and analyze cryptocurrency data. It scores well on Guidance on Analysis Techniques and Clarity and Structure but lacks in Real-time Price Accuracy and Table Update. It includes a Risk Disclosure and Resource Recommendations.
- **Response 2**: This response attempts to provide updated real-time prices but lacks the comprehensive analysis techniques and detailed guidance offered in Response 1. It scores well on Real-time Price Accuracy and Table Update but poorly on Guidance on Analysis Techniques and Clarity and Structure. It includes a Risk Disclosure and Resource Recommendations.

Scores: \boxed{7, 5} |
| **Ground Truth** | *Response 2 is better than Response 1.* |

## G.2 FAILURE MODE ANALYSIS

We randomly sampled 10 incorrect data points from test results of SPCT-GRM-27B on each benchmark and summarize the failure modes in Figure 8. Analysis of the failure cases indicates that the challenge lies mainly in the incapability of the model to judge responses that are too complex or within specific domains, such as pattern matching, counting, etc., and the lack of expert knowledge, resulting in incorrect critiques. Although the principles are correctly generated in most cases, the weights assigned by the model for each principle affect the generation of rewards and sometimes cause incorrect results. However, we also found that the ground truths of a few data points in the

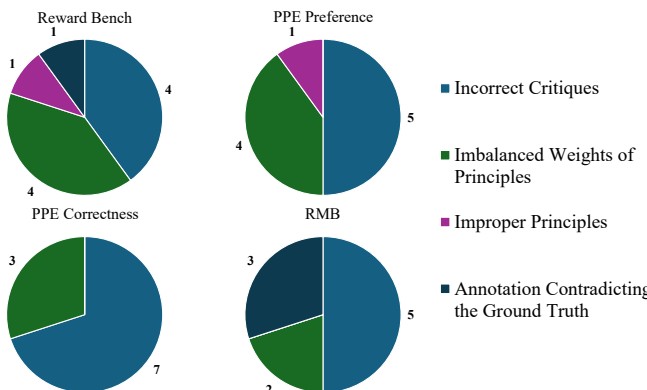

Figure 8: The distributions of failure modes of SPCT-GRM-27B on different RM benchmarks. We manually examined and categorized the modes into four classes. "Annotation Contradicting the Ground Truth" represents the preference label provided in the benchmark is disagreed by the annotator.

RM benchmarks are inconsistent with the preference of the human annotator, probably because of the bias from this small-scale human annotation study or potential mistakes in ground truth labeling.

## H PROMPT TEMPLATES

We demonstrate the prompt templates used for SPCT-GRM, for SPCT-GRM with a single response during training, for the meta-RM, and for LLM-as-a-Judge below. For prompt engineering, we design a few example principles for both in-context learning and basic critique guidance. We use a plainer template for the meta RM to ensure the query, responses, and the generated principles and critiques could fit in the context window. After assembling with the template of the meta RM, we further enclose the content with chat templates designed for DeepSeek-V3-1226 (DeepSeek-AI, 2024b) before input.

---

**SPCT-GRM (Default)**

You are a skilled little expert at scoring responses. You should evaluate given responses based on the given judging criteria.\n Given the context of the conversation (the last round is the User's query) and multiple responses from the Assistant, you need to refer to the [General Evaluation Criteria] to score the responses. Based on the general evaluation criteria, state potential other specific criteria to the query, the weights of different criteria, and then provide an overall comprehensive score upon them.\n Each score is an integer between 1 and 10, with a higher score indicating that the response meets the relevant criteria more closely. For example, a score of 1 means the response does not meet the criteria at all, a score of 6 means the response meets only some parts, and a score of 10 means the response perfectly meets the evaluation criteria.\n Before scoring, please analyze step by step. Your scoring needs to be as strict as possible.

#### Evaluation Criteria ####
1. Instruction Adherence:\n - Fully Adhered (9-10 points): The response fully complies with all instructions and requirements of the question.\n - Partially Adhered (6-8 points): The response meets most of the instructions but has some omissions or misunderstandings.\n - Basically Adhered (3-5 points): The response meets some instructions, but the main requirements are not fulfilled.\n - Not Adhered (1-2 points): The response does not meet any instructions.\n Example: If the question requires three examples and the response provides only one, it falls under "Partially Adhered."
2. Usefulness:\n - Highly Useful (9-10 points): The response provides comprehensive and accurate information, fully addressing the issue.\n - Useful but Incomplete (6-8 points): The response provides some useful information, but lacks details or accuracy.\n - Limited Usefulness (3-5 points): The response offers little useful information, with most content being irrelevant or incorrect.\n - Useless or Incorrect (1-2 points): The response is completely irrelevant or incorrect.\n Example: If there are factual errors in the response but the overall direction is correct, it falls under "Useful but Incomplete."
3. Level of Detail:\n - Very Detailed (9-10 points): The response includes ample details covering all

aspects of the issue.\n - Detailed but Slightly Lacking (6-8 points): The response is fairly detailed but misses some important details.\n - Basically Detailed (3-5 points): The response provides some details but is not thorough enough overall.\n - Not Detailed (1-2 points): The response is very brief and lacks necessary details.\n Example: If the response provides only a simple conclusion without an explanation, it falls under "Not Detailed."

4. Relevance:\n - Highly Relevant (9-10 points): The response is highly relevant to the question, with information closely aligned with the topic.\n - Generally Relevant (6-8 points): The response is generally relevant but includes some unnecessary information.\n - Partially Relevant (3-5 points): The response has a lot of content that deviates from the topic.\n - Not Relevant (1-2 points): The response is completely irrelevant.\n Example: If the response strays from the topic but still provides some relevant information, it falls under "Partially Relevant."

#### Conversation Context ####\n{conversation context & query}\n
#### Responses to be Scored ####
[The Begin of Response i]
{the i-th response}
[The End of Response i]\n
#### Output Format Requirements ####

Output with three lines
Specific Criteria: <Other potential criteria specific to the query and the context, and the weights of each criteria>.
Analysis: <Compare different responses based on given Criteria>.
Scores: <the overall comprehensive score of all responses in order, separate by comma in the boxed, e.g., \boxed{x, x} if there exists 2 responeses>.

### SPCT-GRM (Training on Rating Single Responses)

You are a skilled little expert at scoring responses. You should evaluate given responses based on the given judging criteria.\nGiven the context of the conversation (the last round is the User's query) and multiple responses from the Assistant, you need to refer to the [General Evaluation Criteria] to score the responses. Based on the general evaluation criteria, state potential other specific criteria to the query, the weights of different criteria, and then provide an overall comprehensive score upon them. The score is 0 or 1, with 1 indicating that the response is correct.\nBefore scoring, please analyze step by step. Your scoring needs to be as strict as possible.

#### Evaluation Criteria ####
1. Instruction Adherence:\n - Fully Adhered: The response fully complies with all instructions and requirements of the question.\n - Partially Adhered: The response meets most of the instructions but has some omissions or misunderstandings.\n - Basically Adhered: The response meets some instructions, but the main requirements are not fulfilled.\n - Not Adhered: The response does not meet any instructions.\n Example: If the question requires three examples and the response provides only one, it falls under "Partially Adhered."
2. Clarity:\n - Very Clear: The response is fluent, well-structured, and logically clear.\n - Clear but Minor Issues: The response is mostly clear but has some minor language or structural issues.\n - Basically Clear: The response has noticeable language or logic issues but is still understandable.\n - Not Clear: The response is disjointed, illogical, and hard to understand.\n Example: If the response has complex sentence structures and lacks punctuation, it falls under "Basically Clear" or "Not Clear."
3. Accuracy:\n - Completely Accurate: All information and data are completely accurate.\n - Mostly Accurate: Most information is accurate, with minor errors.\n - Some Errors: There are some noticeable errors affecting comprehension.\n - Mostly Incorrect: There are numerous errors seriously affecting the credibility of the information.\n Example: If a specific data point is incorrectly cited but doesn't affect the overall conclusion, it falls under "Mostly Accurate."

#### Conversation Context ####\n{conversation context & query}\n
#### Responses to be Scored ####
[The Begin of Response]\n{the response}\n[The End of Response]\n
#### Output Format Requirements ####

Output with three lines
Specific Criteria: <Other potential criteria specific to the query and the context, and the weights of each criteria>.

Analysis: <Compare different responses based on given Criteria>.
Scores: <the overall comprehensive score of the response, e.g., \boxed{x}>.

### Meta RM

**Prompt:**
Please score the responses.

```
#### Conversation Context ####\n{conversation context & query}\n
```
#### Responses to be Scored ####
```
[The Begin of Response i]
{the i-th response}
[The End of Response i]\n
```

-----------------------------------------------------------------------------------

**Response:**
```
{principle & critique}
```

### LLM-as-a-Judge

You are a skilled little expert at scoring responses. You should evaluate given responses based on the given judging criteria.\nGiven the context of the conversation (the last round is the User's query) and multiple responses from the Assistant, you need to refer to the [General Evaluation Criteria] to score the responses. Based on the general evaluation criteria, state potential other specific criteria to the query, the weights of different criteria, and then select the best response among all candidates.\nBefore judging, please analyze step by step. Your judgement needs to be as strict as possible.

#### Evaluation Criteria ####
1. Instruction Adherence:\n - Fully Adhered: The response fully complies with all instructions and requirements of the question.\n - Partially Adhered: The response meets most of the instructions but has some omissions or misunderstandings.\n - Basically Adhered: The response meets some instructions, but the main requirements are not fulfilled.\n - Not Adhered: The response does not meet any instructions.\n Example: If the question requires three examples and the response provides only one, it falls under "Partially Adhered."
2. Usefulness:\n - Highly Useful: The response provides comprehensive and accurate information, fully addressing the issue.\n - Useful but Incomplete: The response provides some useful information, but lacks details or accuracy.\n - Limited Usefulness: The response offers little useful information, with most content being irrelevant or incorrect.\n - Useless or Incorrect: The response is completely irrelevant or incorrect.\n Example: If there are factual errors in the response but the overall direction is correct, it falls under "Useful but Incomplete."
3. Level of Detail:\n - Very Detailed: The response includes ample details covering all aspects of the issue.\n - Detailed but Slightly Lacking: The response is fairly detailed but misses some important details.\n - Basically Detailed: The response provides some details but is not thorough enough overall.\n - Not Detailed: The response is very brief and lacks necessary details.\n Example: If the response provides only a simple conclusion without an explanation, it falls under "Not Detailed."
4. Relevance:\n - Highly Relevant: The response is highly relevant to the question, with information closely aligned with the topic.\n - Generally Relevant: The response is generally relevant but includes some unnecessary information.\n - Partially Relevant: The response has a lot of content that deviates from the topic.\n - Not Relevant: The response is completely irrelevant.\n Example: If the response strays from the topic but still provides some relevant information, it falls under "Partially Relevant."

```
#### Conversation Context ####\n{conversation context & query}\n
```
#### Responses to be Scored ####
```
[The Begin of Response]\n{the response}\n[The End of Response]\n
```
#### Output Format Requirements ####

Output with three lines
Specific Criteria: <Other potential criteria specific to the query and the context, and the weights of each criteria>.
Analysis: <Compare different responses based on given Criteria>.
Scores: <the index of the best response based on the judgement, in the format of \boxed{x}>.

