# OpenReview forum: "Inference-Time Scaling for Generalist Reward Modeling"
_ICLR.cc/2026/Conference — Submitted to ICLR 2026_

### Official Review · Reviewer_NHfk · 2025-10-25

**Soundness:** 2
**Presentation:** 2
**Contribution:** 2
**Rating:** 2
**Confidence:** 3

**Summary:**

This paper aims to improve pointwise generative reward modeling (GRM) via: (A) a new training scheme, Self-Principled Critique Tuning (SPCT), consisting of one round of rejective fine-tuning followed by rule-based online RL; and (B) test-time scaling through parallel sampling with voting, guided by a meta reward model that filters low-quality samples. The proposed pipeline reports improvements over comparably sized and larger models on multiple reward-modeling benchmarks.

**Strengths:**

The paper tackles an important problem: improving reward models; which could have substantial impact.

**Weaknesses:**

- **Unclear novelty**. It is not clear whether the contribution is primarily the integration of existing techniques (rejection sampling, GRPO, expert voting) for reward modeling, or whether the fine-tuning and inference-time scaling methods themselves are novel. If the latter, comparisons to closely related methods are missing, making it hard to assess the incremental contribution.

- **Fairness of comparisons**. In Table 2, the method appears to compare fine-tuned reward models with additional inference budget against pre-trained baselines. A fairer comparison would allocate the same fine-tuning and inference budgets to baselines using standard techniques (e.g., GRPO, self-consistency).

- **Limited ablations.** The pipeline has several components (rejective fine-tuning, rule-based RL, voting, meta reward model). Additional ablations are needed to quantify the contribution of each part.

**Questions:**

- The paper repeatedly emphasizes flexibility across input types and lists it as one of four desiderata for reward models. I did not find a concrete justification for this motivation. Can the authors provide additional explanation, examples, and/or literature supporting why input-type flexibility is critical here?
- Section 2.2: How are the filtered principles used in practice? Are principles generated once and reused across questions, or is a new set generated per question?

---

> ### Author Response · Authors · 2025-12-04
>
> We thank the reviewers for their time and for reviewing our manuscript. We have considered the comments raised and have provided our point-by-point responses below.
> ## Contributions
> > (W1) **Unclear novelty**...
>
> As stated in L018-024, we enhance the inference-time scalability of generalist RM in three components:
> 1. For the RM approach, we analyze different approaches and conclude that pointwise GRM is best option.
> 2. For the learning method, we propose SPCT to foster scalable reward generation behaviors in GRMs.
> 3. For the inference-time scaling setting, we use parallel sampling and find a meta RM guidance is effective for SPCT-GRMs.
> ## Experimentation
> > (W2) In Table 2, the method appears to compare fine-tuned reward models with additional inference budget against pre-trained baselines. A fairer comparison would allocate the same fine-tuning and inference budgets to baselines using standard techniques (e.g., GRPO, self-consistency).
>
> In Table 2, the pre-trained models are used for reference. Baseline methods are trained with the same training data (if applicable) to SPCT-GRMs, and uses the same voting numbers for comparisons.
> > (W3) The pipeline has several components (rejective fine-tuning, rule-based RL, voting, meta reward model). Additional ablations are needed to quantify the contribution of each part.
>
> Table 4 presents ablations on different components in RL training, different voting numbers, and the meta RM.
> ## Input Flexibility
> > (Q1) Can the authors provide additional explanation, examples, and/or literature supporting why input-type flexibility is critical here?
>
> In practice, RMs could be required to judge whether a single response is better, and which one of the two or more responses is better, for RL or data annotation.
>
> As shown in Fig.2, pairwise methods are not input flexible. Because, as stated in L1187, "Though the pairwise approach could be extended to" evaluating more than two candidates with extra techniques (e.g., ranking multiple times in tournament sorting, etc.), "it could not be applied to single response scoring." The extra costs in the extra techniques and the obstacles in rating single response make input flexibility important.
> ## Methodology
> > (Q2) Section 2.2: How are the filtered principles used in practice? Are principles generated once and reused across questions, or is a new set generated per question?
>
> In Sec.2.2 and for SPCT-GRMs in other experiments, a new set of principles will be generated for each query.

---

### Official Review · Reviewer_APHk · 2025-10-26

**Soundness:** 3
**Presentation:** 3
**Contribution:** 3
**Rating:** 6
**Confidence:** 5

**Summary:**

This work develops generalist reward models (GRMs) for LLMs that are flexible across various input types (e.g., single, paired, or multiple responses) and scalable at inference time. The authors show that conventional scalar and pairwise RM paradigms often fail on one or both of these fronts.

**Strengths:**

1. The main strength of this work is the combination of the three distinct concepts to build a scalable GRM. The idea of adaptive principle generation trains the GRM to generate principles dynamically based on the specific query and candidate responses, and the model can achieve a more context-aware evaluation. Also, the authors use RL to train the GRM itself, allowing for self-improvement. Finally, the authors show that a Meta RM can guide inference, and the authors introduce a filtering mechanism that improves the reliability of the final aggregated reward.

2. The experimental design is thorough, showing credibility to the paper's claims. The authors experiment on fair baselines by re-implementing alternative methods (e.g., LLM-as-a-Judge, BT-RM, PairRM, CLoud-Gemma) on the same base model (Gemma-2-27B) and using compatible training data. The authors' effort ensures a robust evaluation, since the performance differences are attributable to the method itself, not confounding factors like base model choice or training data.

3. The paper motivates the need for using GRMs. The taxonomy of RM approaches presented in Figure 2 is a contribution in its own right, offering a clear categorization of reward generation paradigms (scalar, semi-scalar, generative) and scoring patterns (pointwise, pairwise).

4. The comparison between inference-time and training-time scaling is also significant. The result that the 27B model with Meta RM-guided voting can outperform a much larger 671B model with greedy decoding provides strong empirical support for the "inference scaling laws" hypothesis.

**Weaknesses:**

1. The central idea of the paper (i.e., training a model to generate natural language principles that explain and reconstruct preference data) is precisely the problem statement in the recent paper Inverse Constitutional AI (Findeis et al., 2024). Their framework explicitly formulates this as a compression task: finding a compact, human-readable "constitution" (a set of principles) that allows an LLM to reproduce the judgments in a large preference dataset. The SPCT method can be understood as a specific, end-to-end algorithmic solution to their problem. In Findeis et al, they use a multi-step pipeline of generation, clustering, testing, and filtering to extract these principles. SPCT's use of RFT and RL is a different approach to the same fundamental goal. The authors do not cite this work, nor discuss, or differentiate themselves from this highly relevant body of work. This overstates the conceptual novelty of Self-Principled Critique Tuning" and misses an opportunity to frame the work as a new, and possibly superior, method for solving the problem.

2. The paper repeatedly describes its second training phase as "rule-based online RL". This is inconsistent with the standard definition of online RL. Online RL involves an agent learning from data collected through direct, on-policy interaction with a dynamic environment. The policy is updated, it collects new data with the new policy, and this loop continues. The method described in this paper, however, uses the GRPO algorithm to fine-tune the GRM on a static, pre-collected preference dataset. The "rollouts" are simply generations from the GRM on prompts from this fixed dataset, and the "reward" is a binary signal based on comparison to a static ground-truth label. This is a form of offline policy gradient fine-tuning. This terminological inaccuracy is not semantic -- it is confusing and mispositions the work relative to research in online RL for reward model training, especially for agentic tasks. For example, recent methods like OPRL/iStar describe frameworks where a reward model is updated concurrently with an agent's policy using newly collected trajectories from an interactive environment. The authors should use a more precise term, such as "offline RL fine-tuning" or "policy gradient optimization on preference data" to describe their method.

3. The paper uses methods to guide critique generation, which is a form of structured reasoning. This connects to a lot of research which focuses on explicitly improving the reasoning capabilities of GRMs to make them better judges. For example, ReasonGRM use a multi-stage framework to generate and filter high-quality chain-of-thought reasoning paths to train a more robust GRM. Other works also showed correlations between an LLM's general reasoning performance and its effectiveness as a generative judge. The principles in this paper are a simplified form of reasoning. The current work focuses entirely on the final outcome (accuracy), neglecting the process (i.e., the principles themselves), which are a key element in this related work.

4. Though the quantitative results are strong, the paper does not have a deep qualitative analysis of its core mechanism: the generated principles. The case studies in the appendix show full model outputs but do not systematically analyze the principles themselves. What kinds of principles does the model learn? How do they evolve from the RFT stage to the final RL-tuned model? Do they differ meaningfully across domains (e.g., coding vs. creative writing)? Answering these questions is important to better understand what the model is learning and whether it has developed a genuinely robust and interpretable set of evaluation criteria.

5. (Minor) I would appreciate if the authors improve the clarity of their figures; in particular Figure 2. This is a really important and useful figure, but it is not completely clear. Please clarify what each method does in the caption for readers that are less familiar with the different approaches, and also explain why some of them are e.g., inference scalable, etc.

**Questions:**

Q1. Could you please elaborate on the relationship between your work and the Inverse Constitutional AI framework? Specifically, how does the SPCT training process differ from or improve upon existing algorithms, which often rely on a "generate cluster filter" pipeline?

Q2. How do the principles generated by the final SPCT-GRM differ from those produced by the initial RFT model? Do they become more abstract, specific, or complex through the RL phase?

Q3. Do the generated principles vary in a structured way across different task domains (e.g., the Reasoning, Safety, and Chat subsets of Reward Bench)?

Q4. Could you provide additional details on the meta RM training process?

Q5. What was the approximate ratio of positive (correctly ranked) to negative (incorrectly ranked) examples in the Meta RM's training set?

Q6. Did you evaluate the Meta RM as a standalone classifier? How well-calibrated are its scores? Does its performance degrade when judging outputs from a policy that drifted significantly from the training data?

---

> ### Author Response · Authors · 2025-12-04
> **Part 1 of 2**
>
> We thank the reviewers for their time and for reviewing our manuscript. We have considered the comments raised and have provided our point-by-point responses below.
> ## Discussion about Related Works
> > (W1) The central idea of the paper is precisely the problem statement in the recent paper Inverse Constitutional AI (Findeis et al., 2024). Their framework explicitly formulates this as a compression task ... misses an opportunity to frame the work as a new, and possibly superior, method for solving the problem.
> > (Q1) Could you please elaborate on the relationship between your work and the Inverse Constitutional AI framework?
>
> [1-3] formulate a task named Inverse Constitutional AI, which requires finding a set of principles to represent a preference dataset so that LLMs could recover the original annotated preference under the guidance of the set of principles. We acknowledge the works are related, and the relations are discussed as follows:
> - **Using SPCT in ICAI tasks:** We appreciate the idea of using SPCT on the ICAI task. However, we do not view it to align with the ICAI objective. The current scope of the paper seeks to optimize reward models to better judge unseen queries or LLM responses. SPCT reinforces both principles and critique generation in GRMs. (1) Since the model is trained on the preference dataset, SPCT-GRM could directly recover the training set to a large extent. (2) Though SPCT-GRM-produced principles have shown transferability (Appendix F.2), the generated principles are optimized specifically for SPCT-GRMs in the objective and could not fit other models to the best. Hence, **we do not claim SPCT a better approach to ICAI tasks**.
> - **Using ICAI approaches in RMs:** ICAI approaches are orthogonal to reward model optimization methods. ICAI approaches could improve RMs by auto-collecting a set of principles on the training data, thus accelerating traditional manual Constitutional AI approaches. In practice, RMs need to judge unseen prompts and responses potentially beyond the fixed principle set. In such cases, a new set of principles or updates on the principle set is needed. As stated in Sec.3.1, SPCT-GRM generates principles according to the given query and candidate responses, avoiding this process.
>
> > (Q1) Specifically, how does the SPCT training process differ from or improve upon existing algorithms, which often rely on a "generate cluster filter" pipeline?
>
> - **Methodology:** As stated in Sec.3.1, SPCT-GRM does not use a fixed set of principles but generates different principles to different queries and responses. It avoids the collection of principles for different datasets.
> ## Analysis on Principles
> > (W4) ... What kinds of principles does the model learn? ... Do they differ meaningfully across domains (e.g., coding vs. creative writing)? ...
> > (Q2) Do the generated principles vary in a structured way across different task domains?
>
> As stated in Appendix F.2, we found that "the principles generated by SPCT-GRM-27B are robust and transferable to other models", e.g., gpt-4o. We did not run large-scale human experiments to check the quality of principles. However, we did qualitatively analyze the generated principles in Appendix G.1 and found it vary in a structured way across different task domains. We also analyzed the failure modes of SPCT-GRM to attribute the errors in Appendix G.2, where principles are correlated in most cases.
> > (W4) ... How do they evolve from the RFT stage to the final RL-tuned model? ...
> > (Q2) How do the principles generated by the final SPCT-GRM differ from those produced by the initial RFT model? Do they become more abstract, specific, or complex through the RL phase?
>
> As stated in Appendix F.4, the generated principles becomes more adaptive after RL compared to the RFT model. We have observed serveral differences including:
> 1. RL models generate more similar and short principles on simple data points (e.g., some safety-related samples), and tend to generate more detailed principles on complex tasks (reasoning), queries with long candidate responses, or queries with multiple candidates.
> 2. RL improves SPCT-GRMs on reasoning-heavy tasks more significantly (Table 8, 9).

---

> ### Author Response · Authors · 2025-12-04
> **Part 2 of 2**
>
> ## Terminology
> > (W2) The paper repeatedly describes its second training phase as "rule-based online RL". This is inconsistent with the standard definition of online RL... The authors should use a more precise term, such as "offline RL fine-tuning" or "policy gradient optimization on preference data" to describe their method.
>
> The difference lies in the definition of the **policy** to be optimized in RL. In this work, the policy is referred only to the reward model and does not incorporate downstream LLMs. The term "online" in this work refers to roll out trajectories from the RM policy based on upcoming training data. And it is off-policy training [4]. If the policy is both the downstream LLM and the reward model, the term "online RL" should feature rolling-outs on both the downstream LLM and the RM.
> ## Relation between Principle Generation and Reasoning
> > (W3) The paper uses methods to guide critique generation, which is a form of structured reasoning. This connects to a lot of research which focuses on explicitly improving the reasoning capabilities of GRMs to make them better judges...
>
> We do view principle generation as a form of structured reasoning. As shown in Fig.4 (b), reasoning capabilities incentivized from reasoning-heavy verifiable tasks do not benefit general RM tasks. As stated in L470, "we do not find an effective way to incentivize long horizon reward generation for generalist reward modeling with general RM data as DeepSeek-R1, and we leave the combination of reasoning and principle-guided reward generation in a single model for future engineering efforts." It is good to see that, as discussed in L467, serveral efforts are made to combine the reasoning abilities into GRMs by prompting, distillation, or mixed training.
> ## Meta RM
> > (Q4) Could you provide additional details on the meta RM training process?
> > (Q5) What was the approximate ratio of positive (correctly ranked) to negative (incorrectly ranked) examples in the Meta RM's training set?
>
> The training data for the meta RM contains approximately same amount of positive and negative samples in total. However, as stated in Sec.4. The data is sampled from SPCT-GRM (or from its RFT data) to represent the policy-specific data distribution. Thus, there are easier prompts with all correct judgement and harder prompts with all incorrect judgement in the training data.
> > (Q6)  Did you evaluate the Meta RM as a standalone classifier? How well-calibrated are its scores? Does its performance degrade when judging outputs from a policy that drifted significantly from the training data?
>
> As stated in L322, the meta RM is trained for SPCT-GRM policy. If the RM policy changes, e.g, removing principle generation, we do not expect or claim the meta RM performs well under that setting.
>
> [1] Findeis et al. Inverse Constitutional AI: Compressing Preferences into Principles. ICLR 2025.
> [2] Kostolansky. Inverse Constitutional AI.
> [3] Kostolansky et al. Iterative Interactive Inverse Constitutional AI.
> [4] Shao et al. DeepSeekMath: Pushing the Limits of Mathematical Reasoning in Open Language Models.

---

### Official Review · Reviewer_6po3 · 2025-10-28

**Soundness:** 3
**Presentation:** 2
**Contribution:** 2
**Rating:** 4
**Confidence:** 4

**Summary:**

- This paper proposes self-principled critique tuning (SPCT), a training procedure aimed at improving the generalist capability of reward models. The two specific goals are to develop generalizable reward models (to evaluate many tasks) and those that scale with test-time compute.
- This is done by combining a rejection sampling and rule-based online RL that aim to 'teach' the model how to generate correct principles for different tasks, and how to derive rewards based on these principles. A meta reward model is also trained to filter out low-quality reward samples before voting.

**Strengths:**

- The paper targets a timely question: of improving reward modeling with scaling test-time budget and generalist capabilities (essentially teaching the RM a system of evaluating through principles); while similar principles are applied to improve reasoning, the focus of this work is on improving RMs (which AFAIK, is under-explored)
- The training recipe (SPCT) is clear and follows established algorithms that have been shown to work (rejection finetuning, online RL with rule-based rewards, and meta-RM). In some sense, the work applies a popular recipe to improving reward models and not reasoning performance
- The ablations indicate that principle guidance, and hinted/non-hinted sampling improve performance and inference time scaling; the empirical results are broad and generally convincing

**Weaknesses:**

- My feeling is that the claimed novelty is somewhat overstated. Almost every building block is known: GRMs that produce critiques; RL-style optimization of LLMs (here RMs); inference-time scaling (through best-of-k, majority voting); and learned verifiers (or here learned verifiers of learned verifiers). This also includes the post-training procedures, e.g., rejection sampling + GRPO-style RL. To me, the main novelty is such techniques are explicitly utilized to optimize for inference-time scalability of reward models. In this sense, the authors should more clearly articulate what is genuinely new and what is more 'systems design'
- The idea of SPCT, of tuning principles, can be explained more clearly in the initial sections. This is a point about clarity only. The high-level idea is quite simple, and it is about improving principle generation/rationalization for generating reward scores; and the method used to achieve this is rejection sampling + RL.
- One limitation of the experiments is that they don't really demonstrate 'generalist' reward modeling in a convincing way; the benchmarks are largely instruction-following / helpfulness / preferences setup. Perhaps I misunderstood, but the paper leans on the 'generalist RM' angle quite heavily.
- The claim that scaling test-time inference > scaling model size during training is fine, but also uncalibrated. For example, sampling N times a 27B model might be more expensive than a single pass of a 200B+ model

**Questions:**

- From an algorithmic standpoint, what are the specific ingredients of SPCT that prior GRM / LLM-as-judge method do not have. Is it the decision to treat principle generation as part of the learned policy? Or the meta-RM guided voting for reward generation
- About MetaRM: The meta reward model is trained as a binary classifier over sampled trajectories, to predict whether a trajectory’s extracted scores match ground truth. This is effectively a “judge of the judge.” How robust is MetaRM to distribution shift (e.g., new domains not in training)?
- It would be good to back up the claim of 27B TTC rivaling 671B with normalized efficiency numbers (e.g., total FLOPS or tokens); it is harder to interpret the practical significance of the scaling claims.

---

> ### Author Response · Authors · 2025-12-04
>
> We thank the reviewers for their time and for reviewing our manuscript. We have considered the comments raised and have provided our point-by-point responses below.
> ## Contributions
> As stated in L018-024, we enhance the inference-time scalability of generalist RM in three components:
> 1. For the RM approach, we analyze different approaches and conclude that pointwise GRM is best option.
> 2. For the learning method, we propose SPCT to foster scalable reward generation behaviors in GRMs.
> 3. For the inference-time scaling setting, we use parallel sampling and find a meta RM guidance is effective for SPCT-GRMs.
>
> > (W1) (Q1) what are the specific ingredients of SPCT that prior GRM / LLM-as-judge method do not have. Is it the decision to treat principle generation as part of the learned policy?
>
> As stated in Sec.3.1, the key difference of SPCT is to treat principle generation as part of the learned policy, which improves inference-time scaling performance (Table 4). The meta RM guidance is effective for SPCT-GRMs to further improve scaling, but it is also orthogonal and could be trained for other GRMs.
> ## Inference-Time Scaling
> > (W4) (Q3) It would be good to back up the claim of 27B TTC rivaling 671B with normalized efficiency numbers (e.g., total FLOPS or tokens); it is harder to interpret the practical significance of the scaling claims.
>
> First of all, as stated in L058-059, we want to emphasize that **inference-time scalability primarily aims to improve overall RM performance regardless of size instead of efficiency**. With inference-time scaling, RMs could achieve unprecedented performance, just like large reasoning models with long chains of thoughts.
>
> Since the 27B model is dense and models in other sizes are MoE, it is difficult to do a completely fair comparison. In terms of costs, we conclude in two dimensions: hosting cost (memory requirements) and inference compute cost (FLOPs).
> 1. **Hosting Cost (The Accessibility Bottleneck):** The hosting cost is primarily determined by the total parameter count. The 671B MoE model requires approximately **25$\times$** more GPU memory (VRAM) to load than the 27B dense model. For users constrained by memory, inference-time scaling on the 27B model provides the only viability path to achieve performance comparable to the 671B model.
> 2. **Inference Compute Cost:** While the 671B model is massive, its MoE architecture activates only \~37B parameters per token. Consequently, the FLOPs for a single forward pass of the 671B MoE are only **\~1.5$\times$** that of the 27B dense model.
> In practical, it is recommended to deploy the largest servable model and leverage inference-time scaling with a meta RM to maximize performance, rather than replacing large models with small ones.
> ## Generalist RM
> > (W3) they don't really demonstrate 'generalist' reward modeling in a convincing way; the benchmarks are largely instruction-following / helpfulness / preferences setup.
>
> As stated in L061, we refer generalist reward modeling to (1) flexibility for different input types and (2) accurate reward generation in various domains.
> - For (1), we consider a generalist RM to support rating single and multiple responses. As stated in Sec.5.1, benchmarks have included rating single or multiple (>2) candidates.
> - For (2), we consider a generalist RM to have a general training objective without constraints in specific domains, e.g., domains with or without groundtruths, etc. As stated in Sec.5.1 and Appendix E.2, we select large-scale benchmarks covering helpfulness (chatting, assisting, writing, role playing, error detecting, etc.), safety/harmlessness (crime, sex, privacy, harm, etc.) and reasoning-heavy verifiable tasks (code, math, instruction following, scientific QA, etc.) from crowdsourcing and semi-auto collecting on synthetic or real-world queries. We think RMs, as human proxies, deliver preference signals. We acknowledge the benchmarks do not cover all domains about preference, but we think SPCT could be applied to broader RM tasks in recent benchmarks, e.g., agentic web browsing [1].
> ## Meta RM
> > (Q2) How robust is MetaRM to distribution shift (e.g., new domains not in training)?
>
> Distribution shifts on the meta RM could be interpreted in two ways:
> 1. **Shifts in the RM policy:** As stated in L322, the meta RM is trained for SPCT-GRM policy. If the RM policy changes, e.g, removing principle generation, we do not expect the meta RM performs well under that setting.
> 2. **Shifts in the RM task domain:** As stated in Sec.4, the training data of meta RM does not overlap with the test data. We found the meta RM could better remove low-quality judgement of SPCT-GRM instead of directly picking the best one. As shown in Table 4, when applying $k_{\text{meta}}=1$, the inference-time scaling performance is increased but still not as good as larger $k_{\text{meta}}$.
>
> [1] Lù et al. AgentRewardBench: Evaluating Automatic Evaluations of Web Agent Trajectories.

---

### Official Review · Reviewer_6ygt · 2025-11-01

**Soundness:** 2
**Presentation:** 2
**Contribution:** 2
**Rating:** 4
**Confidence:** 3

**Summary:**

This paper addresses the issue of reward modeling for RL post-training for LLMs. It tackles this problem through introducing a technique called self-prinicipled critique tuning (SPCT) for generalist reward modelling (GRM). Essentailly, SPCT first gets the model to generate its own criteria, specific to the task, then uses those criteria to generate a score.

Authors also did experiments with a model that is finetuned with this technique (SPCT-GRM-27B) to show SPCT doing well in various reward benchmarks, and also being scalable at inference-time through using muiltiple of such reward models and voting.

**Strengths:**

- Solid engineering work that could potentially be useful / scaleable for real-life scenarios.
- Rigorous testing methods with reproduction of baselines from other papers.
- The authors provide a fair towards its own work also, e.g. The authors highlight other models performing better on benchmarks, and showing failure modes in Appendix.

**Weaknesses:**

- Limited novelty - there are numerous papers in "LLM self generating criteria for further scoring"
- Limited metrics on inference - The authors mentioned that 32x SPCT-GRM matches a 671B model in performance. However, is the inference time faster / compute requirement less?
- Performance - Is the performance gain mostly from scaling voting / the meta-RM model? If so, does the SPCT component actually matter?
- Potential data contamination issue? e.g. I noted that the authors finetuned its model on several datasets including Skywork-80k, which has overlap with RB (https://huggingface.co/datasets/natolambert/skyworks-rewardbench-contamination)

**Questions:**

- Has the author explored whether your implementation of RM improves finetuning performance? Would be interesting to see whether benchmark performance translates to real-life performance
- To address the potential data contamination issue above, has the author looked into zero-shot versions of SPCT without finetuning?
- In addition, it seems that the meta-reward model is doing a lot of heavy lifting. Has the authors looked into how / why that is the case?
- Any analysis on whether the principles generated actually correlate to the task?
- Additionally, did the authors look into how the decision boundary / score distributions of the reward models look like?

---

> ### Author Response · Authors · 2025-12-04
>
> We thank the reviewers for their time and for reviewing our manuscript. We have considered the comments raised and have provided our point-by-point responses below.
> ## Data Contamination Issues
> > (W4) ...Skywork-80k, which has overlap with RB
> > (Q2) To address the potential data contamination issue above, ...
>
> As stated in L1238, we use [Skywork-Reward-Preference-80K-v0.2](https://huggingface.co/datasets/Skywork/Skywork-Reward-Preference-80K-v0.2) in the training data, which, to the best of our knowledge, does not have data contamination issues. We have also checked other datasets.
> ## Inference-Time Scaling
> First of all, as stated in L058-059, we want to emphasize that **inference-time scalability primarily aims to improve overall RM performance regardless of size instead of efficiency**. With inference-time scaling, RMs could achieve unprecedented performance, just like large reasoning models with long chains of thoughts.
>
> > (W2) 32x SPCT-GRM matches a 671B model in performance. However, is the inference time faster / compute requirement less?
>
> As shown in Fig.4, 32 votes or 4 votes with the meta RM from SPCT-GRM-27B could match/outperform SPCT-GRM-671B. Given that the 27B model is dense and the 671B model is MoE, the comparison on costs is not a direct number division.
>
> In terms of costs, there are two dimensions: hosting cost (memory requirements) and inference compute cost (FLOPs).
> 1. **Hosting Cost (The Accessibility Bottleneck):** The hosting cost is primarily determined by the total parameter count. The 671B MoE model requires approximately **25$\times$** more GPU memory (VRAM) to load than the 27B dense model. For users constrained by memory, inference-time scaling on the 27B model provides the only viability path to achieve performance comparable to the 671B model.
> 2. **Inference Compute Cost:** While the 671B model is massive, its MoE architecture activates only \~37B parameters per token. Consequently, the FLOPs for a single forward pass of the 671B MoE are only **\~1.5$\times$** that of the 27B dense model.
>
> It is recommended to deploy the largest servable model and leverage inference-time scaling with a meta RM to maximize performance, rather than replacing large models with small ones.
> ## Experimental Results
> > (W3) Is the performance gain mostly from scaling voting / the meta-RM model? If so, does the SPCT component actually matter?
>
> SPCT matters in both RM training and inference-time scaling, and is orthogonal to voting numbers and the meta RM.
>
> As shown in ablation studies (Table 4), optimized principles brings better inference-time scaling performance. Specifically, it helps inference-time scaling on more difficult test samples.
> > (Q3) In addition, it seems that the meta-reward model is doing a lot of heavy lifting. Has the authors looked into how / why that is the case?
>
> Meta RM is an important component in inference-time scaling. As suggested in L315, "a few generated samples might be biased or low-quality due to randomness or model limitations."
>
> From our observation, applying the meta RM on a smaller amount of votes (e.g., voting@4) helps remove random low-quality judgements and results in a large performance increase, as shown in Fig.4. As the number of votes goes up, the affect of randomness becomes less. At that point, the meta RM contributes mostly on reasoning-heavy problems which SPCT-GRM does not good at, as shown in Table 2 and Table 7.
> > (Q1) Has the author explored whether your implementation of RM improves finetuning performance?
>
> We use RM benchmarks that are strongly correlated to downstream fine-tuning performance, e.g., PPE [1], instead of performing large-scale downstream RL. Upcoming works (e.g., [2]) and practical services (e.g., [2]) have adopted and shown the merits of using GRMs in RL.
> ## Analysis on Principles and Reward Scores
> > (Q4) whether the principles generated actually correlate to the task?
>
> As stated in Appendix F.2, we found that "the principles generated by SPCT-GRM-27B are robust and transferable to other models", e.g., gpt-4o. We did not run large-scale human experiments to check the quality of principles. However, we did analyze the failure modes of SPCT-GRM to attribute the errors in Appendix G.2, where principles are correlated in most cases.
> > (Q5) how the decision boundary / score distributions of the reward models look like?
>
> Scores generated by the GRMs in a single pass show quite a large variance due to the diversity of generation. However, we observed that:
> 1. Models tend not to rate scores that are all below 5, e.g., {2, 3}, {1, 4, 2}. It might be because the RFT data contains more of high scores due to the model alignment.
> 2. The boundary of highest and lowest scores stabilizes when sampling numbers increase.
>
> [1] Frick et al. How to evaluate reward models for RLHF. ICLR 2025.
> [2] Hone et al. Think-RM: Enabling Long-Horizon Reasoning in Generative Reward Models. NIPS 2025.
> [3] https://platform.openai.com/docs/guides/graders .

---

### Meta-Review · Area_Chair_ZNgC · 2025-12-22

**Summary:**

The primary criticisms centered on limited novelty, as the core components (e.g., self-generating principles, online RL, inference-time voting) were seen as a recombination of existing techniques rather than a fundamental innovation. Questions were raised about the fairness and completeness of experimental comparisons, specifically whether baselines were given equivalent fine-tuning and inference budgets. The practical significance of the inference-time scaling claims was questioned, with requests for more normalized efficiency metrics (e.g., FLOPS) to compare against simply using a larger model.

**Reviewer Concerns:**

The rebuttal addressed some concerns regarding experimental rigor and the role of specific components. It clarified that baselines were trained on the same data and that ablation studies (Table 4) demonstrate the contribution of the SPCT method and meta-RM.
Outstanding concerns primarily relate to the paper's core claims of novelty. The rebuttal frames the contribution as a synergistic integration of three components but does not fully counter the critique that each building block is well-established. The need for a deeper, more systematic qualitative analysis remains unaddressed.

**Reviewer Scores:**

- Reviewer 6ygt (Rating: 4): Given the concerns about novelty and data contamination were only partially alleviated. This reviewer would likely maintain their score of 4. The rebuttal didn't provide a strong reason for a significant upward change.
- Reviewer 6po3 (Rating: 4): This reviewer's primary concern was also novelty. This reviewer might move to a weak 5, but would probably maintain a 4 due to the outstanding novelty question.
- Reviewer APHk (Rating: 6): This reviewer raised points about novelty (Inverse Constitutional AI) and terminology. The rebuttal's detailed engagement with these points, especially the discussion of related work, likely increased their confidence in the paper's scholarly rigor. This reviewer would likely maintain the score.
- Reviewer NHfk (Rating: 2): This reviewer had the strongest criticisms, citing "unclear novelty" and "unfair comparisons." The rebuttal's explanations about baseline training and ablations might have addressed the fairness concern to some degree. However, given the starkly negative initial rating and the fact that the core novelty concern remains, any change would be minor.

---

### Decision · Program_Chairs · 2026-01-26

Reject